



# Systematic overestimation of evapotranspiration over irrigated areas by an offline land surface model

Tanguy Lunel[1], Belén Martí[1], Aaron Boone[1], and Patrick Le Moigne[1]

[1]Centre National de Recherches Météorologiques, Université de Toulouse, Météo-France, CNRS, Toulouse, France

**Correspondence:** Tanguy Lunel (tanguy.lunel@umr-cnrm.fr)

**Abstract.**

Offline Land-Surface Models (LSMs) are essential for a wide range of applications, including water resource management and agricultural planning. A critical variable in these models is evapotranspiration, but its value is easily biased in irrigated areas. In fact, irrigation fundamentally alters local atmospheric conditions - cooling and humidifying the air and reducing wind speeds - factors that contribute to reducing evapotranspiration rates. This phenomenon is called "atmospheric feedback", but is often missing or poorly represented in offline LSMs because most of the atmospheric forcings used, such as reanalyses and climate model outputs, overlook the atmospheric effect of irrigation. This leads to a tendency for offline LSMs to overestimate evapotranspiration rates over irrigated areas. In this study, the atmospheric effects of irrigation are quantified using data from the Land surface Interactions with the Atmosphere over the Iberian Semi-arid Environment (LIAISE) project field campaign. The various surface processes that influence the dynamics of evapotranspiration in response to the atmospheric feedback are then systematically investigated. The results confirm the importance of considering the atmospheric feedback in the Interaction Sol-Biosphère-Atmosphère (ISBA) LSM over irrigated areas in many configurations. For well irrigated crops, the average overestimation of evapotranspiration is about 25%. Conversely, for water-stressed crops, this overestimation is mitigated because the timing of stomatal closure is influenced by atmospheric feedback mechanisms, providing a compensatory effect. These findings highlight the need for improved representation of irrigation-related atmospheric feedback in the atmospheric forcings used as upper boundary conditions in LSMs to improve the accuracy of evapotranspiration estimates in agricultural or hydrological contexts.

**Highlights:**

- Air cooling, humidification, and wind attenuation from irrigation are quantified for two summer weeks.

- Standard offline land surface models overestimate evapotranspiration over irrigated areas.

- The overestimation depends on the model configuration and ranges from 4% to 35%.

- For water-stressed vegetation, stomatal closure reduces the overestimation of evapotranspiration.

**Keywords.** Surface–Atmosphere Interactions, Irrigation, Evapotranspirative demand, Model bias, Land–Atmosphere feedback,



## 1   Introduction

Land surface modelling is increasingly important for weather forecasting, climate projections, water resource management, agricultural systems monitoring and, more generally, for assessing the sustainability of our societies. It can be used to improve our understanding of the behaviour of the land surface, atmosphere and hydrology in response to various natural or human modifications of the environment. Evapotranspiration is at the crossroads of many research fields, and proper modelling of its value is key for understanding and predicting aspects such as soil moisture, surface fluxes, and atmospheric dynamics. The representation of evapotranspiration is a focal point in LSMs as it couples the energy, water and carbon cycles in such schemes. It is defined as the flux of water vapour going from the land surface to the atmosphere. Modelling evapotranspiration in response to natural factors is now fairly well understood and represented in LSMs, but this is not yet the case for anthropogenic influences.

Among the various human interventions that strongly influence evapotranspiration, irrigation is one of the most significant. Irrigation extracts water from rivers or groundwater to make it available to crops, which then release water to the atmosphere through evapotranspiration. In weather and climate science, the effect of irrigation on the atmosphere is an active area of research. In particular, irrigation has been shown to dramatically increase crop evapotranspiration, which then leads to cooling and humidification of the near-surface air (Jochum et al., 2006; Lobell et al., 2008; Sorooshian et al., 2011; Cook et al., 2015; Valmassoi et al., 2020; Lawston et al., 2020; Mcdermid et al., 2023; Lunel et al., 2024a). Irrigation can also affect wind regimes by reducing near-surface wind speed (Sorooshian et al., 2011; Sridhar, 2013; Lunel et al., 2024a) or by affecting the dynamics of mesoscale winds (Phillips et al., 2022; Lunel et al., 2024a, b). Irrigation impacts air temperature, humidity, and wind speed, especially in the lower atmosphere, which, in turn, creates a feedback where changes in the irrigation-altered environment affect surface evapotranspiration. This feedback is hereafter referred to as atmospheric feedback.

Most of the recent results on the influence of irrigation on the atmosphere were made possible by surface–atmosphere coupled models, typically run with and without irrigation (Kueppers et al., 2007; Sorooshian et al., 2011; Cook et al., 2015; Lawston et al., 2020; Lunel et al., 2024a). Such coupled models consistently represent the effect of the irrigated surface on the atmosphere and, conversely, of the influenced atmosphere on the surface, i.e. the atmospheric feedback. However, this is not the case for offline (non-coupled) LSMs. In spatially distributed offline LSMs applications, the atmospheric conditions are provided as input upper boundary conditions, which typically come from weather reanalysis or from General Circulation Model (GCM) outputs, both of which use surface–atmosphere coupled models to produce such data. However irrigation is rarely represented in these coupled models and therefore most meteorological data used in offline LSMs do not consider the effect of irrigation on the atmosphere. The atmospheric conditions resulting from these meteorological forcings are therefore often too warm, too dry, and too windy over irrigated areas (Tuinenburg and de Vries, 2017; Qian et al., 2020). Offline LSMs become biased when the atmospheric feedback is not taken into account. This is the case, for example, in many hydrological and agronomic studies which use atmospheric forcings that do not account for irrigation effects (Rosenberg et al., 2003; Woznicki et al., 2015; Nechifor and Winning, 2019; Gorguner and Kavvas, 2020).



To the best of the authors' knowledge, only Decker et al. (2017) focused on quantifying the atmospheric feedback on evapotranspiration. The authors used the Community Atmosphere Biosphere Land Exchange (CABLE) LSM coupled to the Weather Research and Forecasting (WRF) atmospheric model to investigate the importance of land–atmosphere coupling over irrigated areas in southeastern Australia. They ran two ensembles of six members, with and without an irrigation parameterization, at a horizontal resolution of 10 km. This allowed them to attribute a 0.5$^o$C decrease in 2-m air temperature and a 0.2 to 0.5 g kg$^{-1}$ increase in specific humidity (a 5% to 10% increase in relative humidity) over irrigated land owing to irrigation. The cooler and more humid atmosphere was then shown to reduce the evapotranspiration over irrigated land. Ignoring this effect led to a 25% overestimation of evapotranspiration, and therefore to an equal overestimation of irrigation demand to compensate for evapotranspiration water losses. The authors attributed this effect to the higher humidity rate rather than to the temperature decrease. However, they did not investigate in detail the processes involved in the atmospheric feedback and did not distinguish between evaporation and transpiration. Decker et al. (2017) used a single LSM configuration and provided limited information on the specific setup used to model evapotranspiration. In order to gain a comprehensive understanding of the potential biases that may occur in other LSMs, it is essential to explore a diverse range of evapotranspiration behaviours, i.e. diverse configurations. This can be achieved by varying the parameterizations within a LSM, allowing a more nuanced analysis of different evapotranspiration responses.

One reason why the atmospheric feedback on evapotranspiration has not been studied much is that the effect of irrigation on the atmosphere is often not well characterized. Until now, few studies have been able to clearly attribute some atmospheric changes to irrigation. The LIAISE campaign provides the data to take a step towards addressing this issue. The LIAISE project is an international research endeavor aimed at improving the understanding of natural and anthropogenic land surface processes and their subsequent interactions with the Mediterranean Atmospheric Boundary Layer (ABL) and the hydrological cycle at the basin scale (Boone et al., 2024). The LIAISE field campaign took place in 2021 in northeastern Spain within the Ebro basin east of the city of Lleida. This location was chosen because it includes a heavily irrigated area next to a naturally semi-arid area. The LIAISE Special Observation Period (SOP) ranged from July 15 to July 29, 2021.

Using data from the LIAISE campaign, Brooke et al. (2024) showed that surface energy partitioning was strongly modified by irrigation, with evapotranspiration increasing by a factor of 10 at the irrigated site compared to the rainfed site. This different energy partitioning results in different vertical ABL structures over the irrigated and rainfed sites. In particular, it is shown that the potential temperature between 0 and 800 m a.g.l. is systematically lower over the irrigated site than over the rainfed site, up to -6°C. Lunel et al. (2024a) used a surface–atmosphere coupled model with and without an irrigation parameterization for a LIAISE case study and confirmed that the observations of Brooke et al. (2024) were directly due to irrigation. The authors showed that irrigation induced a mean cooling of -4.7°C and a humidifying effect of +3.4 g kg$^{-1}$ on the near-surface atmosphere over the irrigated area during two specific days of the campaign. The near-surface wind speed was also shown to be reduced by the irrigation, and more generally the ABL circulations were modified. In particular, a breeze circulation between the irrigated and rainfed zones could be attributed to the strong surface thermal contrasts induced by the irrigation. More generally, Lunel et al. (2024a) showed that the coupled model SURFace EXternalisée (SURFEX)–Meso-NH is able to represent well the effects of irrigation on the atmosphere. Using the same approach of combining model and observations,



Lunel et al. (2024b) studied how a local wind called the Marinada was affected by irrigation. The authors found that irrigation was responsible for a general wind speed reduction in the irrigated area and for a delay in the arrival of Marinada relative to a
simulation without irrigation. Udina et al. (2024) investigated the effect of irrigation on precipitation during the SOP using the WRF model. The authors found that irrigation causes a decrease in the atmospheric boundary layer height and in the lifting condensation level. These two effects have opposite effects on the chances of cloud formation and precipitation, and the authors were unable to show a clear effect of irrigation on cloudiness or precipitation during the LIAISE campaign. They attribute this lack of effect on clouds and precipitation to the fact that the perturbed weather situations were driven more by the synoptic
scale than by local processes.

Mangan et al. (2023) and González-armas et al. (2024) investigated the drivers of evapotranspiration fluxes during the LIAISE campaign. Mangan et al. (2023) used the unicolumn simplified model Chemistry Land-surface Atmosphere Soil Slab (CLASS) (Vilà-Guerau De Arellano et al., 2015) and local observations to quantify the tendency terms of evapotranspiration. The authors found that the temporal evolution of evapotranspiration is determined to the first order by the incident shortwave
radiation, then to the second order by the aerodynamic and surface resistances $R_a$ and $R_s$. The aerodynamic resistance $R_a$ depends directly on the wind speed and stability, the latter being a function of wind, roughness, temperature and humidity stratification in the surface layer. González-armas et al. (2024) confirmed the findings of Mangan et al. (2023) by perturbing some parameters in the CLASS model and studying its response in detail. The authors found that photosynthetically active radiation, which is strongly correlated to the incident shortwave radiation, is the primary driver of evapotranspiration, followed
by air temperature and vapour pressure deficit. Although carbon dioxide levels also have an influence, the authors found it to be the least important contributor. Note that although González-armas et al. (2024) did not study the effect of wind, the authors mentioned the fact that its role is also important to consider to fully understand plant evapotranspiration during LIAISE.

Since the LIAISE campaign provides a novel context for which the drivers of evapotranspiration have been well characterized (Mangan et al., 2023; González-armas et al., 2024), and the influence of irrigation on the atmosphere well quantified (Brooke
et al., 2024; Lunel et al., 2024a, b), this campaign also provides a unique framework to study and better understand the various aspects of atmospheric feedback on evapotranspiration. The purpose of this article is to quantify this atmospheric feedback in the ISBA LSM with various configurations in the context of the LIAISE campaign. This study consists in a model sensitivity experiment, focusing on mid-summer atmospheric conditions, when the potential evapotranspiration is at a maximum. It also aims to better understand the LSM response to the atmospheric feedback.

First, the SOP of the LIAISE campaign is modelled using the coupled SURFEX–Meso-NH model with and without irrigation. The results of the two runs are used to quantify the average influence of irrigation on near-surface temperature, humidity and wind during the two weeks of the LIAISE SOP. Second, the atmospheric conditions of the two coupled runs are converted into two atmospheric forcing dataset, which are used in point-scale offline ISBA runs. The thorough study of the difference in modelled evaporation and transpiration resulting from the use of one atmospheric forcing over the other and with different
ISBA configurations allows an improved understanding of the surface processes involved in the atmospheric feedback on evapotranspiration. Finally, the atmospheric feedback is quantified globally for different ISBA configurations during the LIAISE SOP.





## 2 Materials and Methods

### 2.1 ISBA configuration

The LSM used in this study is ISBA (Noilhan and Planton, 1989; Noilhan and Mahfouf, 1996). It is fully integrated into the SURFEX platform which is coupled to the Meso-NH atmospheric model. SURFEX is a collaborative platform software package maintained and developed at Météo-France in partnership with other international collaborators, and version 8.1 is used in this work (Masson et al., 2013). ISBA models different types of natural continental land surface cover, from bare ground to forest, including crops, grasslands and glaciers. It offers a wide variety of options for modelling surface processes:

the options retained for the present work are discussed below and summarized in Table 1 and Table 2. The options not discussed here are left at their default values as given in the SURFEX v8.1 User Guide (Centre National de Recherches Météorologiques, 2020).

The ISBA configuration used in coupled mode with the atmosphere to produce the atmospheric forcings is described in Lunel et al. (2024b). The ISBA configuration used for the point scale offline simulations is similar to that used in the coupled

mode, with some differences due to the fact that the surface features are not taken from a specific land cover database. The point scale offline simulations do not aim to represent a specific field of the LIAISE campaign, but rather an average typical field of the Urgell region.Sand and clay contents are both set at 33%, making it a loamy clay soil. The Leaf Area Index (LAI) is set to 3 m$^2$ m$^{-2}$ in order to represent a well-established crop. The roughness length is set to 0.1 m, which is an average value for irrigated crops such as corn and alfalfa (Jacobs and Van Boxel, 1988; Otsuki et al., 1999). Albedo values for the Ultra-Violet

(UV), VISible (VIS), and Near Infra-Red (NIR) spectral bands are derived from the land cover database ECOCLIMAP-II (Faroux et al., 2013) for land cover type 527, labeled "Spanish Irrigated Crops," as described in the SURFEX/ECOCLIMAP-II User's Guide Centre National de Recherches Météorologiques (2020). This ISBA configuration has 14 vertical soil layers extending from the surface to a depth of 12 meters (Decharme et al., 2013). Soil hydrological parameters are determined using a pedotransfer function based on Cosby et al. (1984) using sand and clay contents. Soil heat and liquid water transfer are

modelled using Fourier and Darcy laws, respectively (Decharme et al., 2011). The heat transfer is modelled explicitly over the entire 12 m soil column, while water transfer is modelled explicitly down to the deepest root depth. Below this depth, water is assumed to drain to the aquifer and no capillary rise from the aquifer is considered in the current study.

### 2.2 Evapotranspiration in ISBA

ISBA models evapotranspiration by first modelling the vapour fluxes from the vegetation and from the bare ground separately.

The vapour flux from the bare ground, $E_g$, is given by Eq. (1).

$$E_g = \frac{\rho_a}{R_a} \left[ hu\, q_{sat}\left(T_s\right) - q_a \right] \tag{1}$$

where $\rho_a$ is the air density $[kg\,m^{-3}]$, $R_a$ is the aerodynamic resistance at the ground level $[s\,m^{-1}]$, $hu$ is the relative humidity at the ground surface $[-]$, $q_a$ is the air specific humidity $[kg\,kg^{-1}]$, $T_s$ is the surface temperature $[K]$, and $q_{sat}(T_s)$ is the saturated specific humidity at the surface $[kg\,kg^{-1}]$.



**Table 1.** Table of fixed ISBA parameters across the study.

| Parameters | Value | Reference |
|---|---|---|
| Sand fraction | 33% | default value |
| Clay fraction | 33% | default value |
| LAI | 3 m m$^{-2}$ | arbitrary value |
| Vegetation fraction | 0.83 | derived from LAI |
| $z_0$ | 0.1 m | arbitrary value |
| Root depth | 66% in the top 40 cm | Jackson et al. (1996) |
| Vegetation albedo UV | 0.06 | ECOCLIMAP-II |
| Vegetation albedo VIS | 0.06 | ECOCLIMAP-II |
| Vegetation albedo NIR | 0.31 | ECOCLIMAP-II |
| Soil albedo UV | 0.1 | ECOCLIMAP-II |
| Soil albedo VIS | 0.1 | ECOCLIMAP-II |
| Soil albedo NIR | 0.15 | ECOCLIMAP-II |
| Water and heat transfer | Diffusive scheme | Decharme et al. (2011) |
| Soil layering | 14 layers | Decharme et al. (2013) |

The water vapour flux from vegetation can be divided into a vapour flux coming from the evaporation of the intercepted water on the leaves and another one coming from the transpiration of the plant. The evaporation of the intercepted water is given by an equation very similar to Eq. (1) with $hu$=1, and the dynamics of the intercepted water vapour flux is therefore similar to $E_g$. However, in the LIAISE context, there is very little rainfall and the fields are flood irrigated, so the vegetation is very rarely covered by intercepted water. The vapour flux from the evaporation of the intercepted water is very small and

is assumed to be zero in the present work to simplify the equation set. Thus, the vegetation vapour flux comes only from the transpiration process and is given by Eq. (2).

$$E_{tr} = \frac{\rho_a}{R_a + R_s} \left[ q_{sat}(T_s) - q_a \right] \tag{2}$$

where $R_a$ is the aerodynamic resistance, $R_s$ is the surface resistance, $q_a$ is the specific humidity of the air, $T_s$ is the surface temperature, and $q_{sat}(T_s)$ is the saturated specific humidity at the surface. The surface resistance $R_s$ represents the response

of the plant to external conditions. The aerodynamic resistances $R_a$ take into account the effect of turbulence, atmospheric stability and wind speed on evaporation or transpiration. It is defined as

$$R_a = \frac{1}{C_E V_a} \tag{3}$$

where $V_a$ is the wind speed, and $C_E$ is the turbulent transfer coefficient. It includes the effect of atmospheric stability and is calculated according to Le Moigne et al. (2018).

Equations 1-3 are used to calculate transpiration and evaporation. The dependence of the vapour fluxes on humidity $q_a$ and wind speed $V_a$ is quite straightforward in the equations. The dependence on air temperature is embedded in the surface



**Table 2.** Table of ISBA options tested for in the present study.

| Parameters | Description | Reference |
|---|---|---|
| Canopy representation | composite approach | Noilhan and Planton (1989) |
| | TSEB approach | Boone et al. (2017); Napoly et al. (2017) |
| Stomatal conductance scheme | ISBA-A-$g_s$ | Jacobs et al. (1996); Calvet et al. (1998) |
| | Jarvis | Jarvis (1976); Noilhan and Planton (1989) |
| Drought response | drought avoidant | Calvet et al. (2004) |
| | drought tolerant | Calvet et al. (2004) |
| Soil moisture | No Irrigation | default value |
| | Fixed SWI = 0 to 1.2 | Not Applicable (NA) |
| | Threshold-based irrigation | Le Moigne et al. (2018) |

temperature $T_s$ and the turbulent transfer coefficient $C_E$, since both vary with the overlying air temperature. These are the variables directly affected by the atmospheric feedback. However, the vapour fluxes also depend on the parameterization options used to model $R_s$ and the global surface modelling approach, which determine how the canopy and the ground interact.

These different ISBA options are key to evapotranspiration modelling and different values are therefore tested throughout the study to highlight how evapotranspiration is affected by these choices. This also ensures that the present conclusions are not limited to specific configurations. These varying ISBA options are summarized in Table 2 and described in more detail below.

### 2.2.1   Canopy representation

The modelling approach used for the vegetation canopy is an important feature that influences how the soil and vegetation
interact. Depending on the approach used, $T_s$, $q_a$, $R_a$ will evolve differently and ultimately influence evapotranspiration. In many LSMs, the strategy for modelling the vegetation canopy is to include it in the top surface layer which is then referred to as the composite layer because it mixes soil and vegetation properties (Noilhan and Planton, 1989). This composite approach considers only one temperature and soil moisture for the composite layer, with properties such as thermal inertia or conductivity that may vary depending on the proportion of vegetation in the composite layer. There is only one surface temperature $T_s$ and
the atmospheric environment is the same for the vegetation and the ground.The total evapotranspiration, $ET$, is then the sum of the vapour fluxes from the vegetation transpiration $E_{tr}$ and the bare ground $E_g$:

$$ET = f_{veg}E_{tr} + (1 - f_{veg})E_g \qquad (4)$$

where $f_{veg}$ is the fraction of vegetation covering the surface. This composite approach is still used in many Numerical Weather Prediction (NWP) systems due to its simplicity and low computational cost (Giard and Bazile, 2000; European Centre for
Medium-Range Weather Forecasts, 2015). However, this simplicity comes at the expense of realism, and the composite approach may yield unrealistic amounts of bare ground evaporation or overestimate the surface soil temperature at high LAI (Napoly, 2016).





The strategy to overcome this issue is to separate the modelling of the soil and of the vegetation (Niu et al., 2011; Best et al., 2011; Napoly, 2016). This strategy is often called Two-Source Energy Balance (TSEB) in LSMs and is called Multi-Energy Balance (MEB) in ISBA (Napoly, 2016; Boone et al., 2017; Napoly et al., 2017) since it also potentially includes a separate snow surface energy budget. For the current study, it computes two separate Surface Energy Balance (SEB) for the ground and the vegetation, the atmospheric properties are discretized in the vertical in the canopy, meaning that $R_a$ and $q_a$ are computed separately for the canopy (giving $R_{a-c}$ and $q_{a,c}$) and for the ground ($R_{c-g}$ and $q_{a,g}$), and the temperature of the canopy and the ground are resolved separately, giving $T_{s,c}$ and $T_{s,g}$ instead of a single surface temperature $T_s$. Transpiration $E_{tr}$ and bare ground evaporation $E_g$ both participate in modifying the atmospheric properties at the canopy level, and total evapotranspiration is then calculated as the vapour flux between this canopy atmosphere and the overlying atmosphere. The full set of equations used in MEB is described in Boone et al. (2017), but is not detailed here for the sake of brevity.

### 2.2.2 Stomatal conductance scheme

The surface resistance $R_s$ (Eq. (2)) is a key variable for modelling the response of the plant to external conditions. This surface resistance is actually mainly determined by the aperture of the leaf stomata and is therefore often referred to as stomatal resistance. It will be referred to as such in the remainder of this work. Stomatal conductance $g_s$, the inverse of stomatal resistance, is also often used in the literature. Stomatal conductance $g_s$ can be modelled using several approaches, two of which are used in ISBA: Jarvis (Jarvis, 1976; Noilhan and Planton, 1989) and A-$g_s$ schemes (Jacobs et al., 1996; Calvet et al., 1998).

The Jarvis parameterization is rather simple and aims at modelling only vapour fluxes. Stomatal conductance in this case is mainly a function of incoming radiation, soil water stress, Vapor Pressure Deficit (VPD), air temperature and a few plant characteristics. In the Jarvis parameterization, the influence of each parameter is independent of the others.

The A-$g_s$ schemes are more recent parameterizations that model the photosynthetic and transpiration processes more realistically, based on the work of Jacobs et al. (1996). It allows the carbon exchange between vegetation and the atmosphere to be modelled explicitly, as well as the interdependence between the parameters already considered in the Jarvis scheme. It has been adapted to ISBA in the so-called ISBA-A-$g_s$ scheme by Calvet et al. (1998). The Jarvis and A-$g_s$ stomatal conductance parameterizations are common to many LSMs (Henderson-Sellers et al., 1993; Le Moigne et al., 2018; Boussetta et al., 2021; Oliver et al., 2022).

### 2.2.3 Drought response

Irrespective of the stomatal conductance parameterization chosen, vegetation characteristics also play a role in determining the stomatal conductance, $g_s$. In particular, the response of plants to water stress is an important feature to consider in ISBA. Calvet (2000) and Calvet et al. (2004) identified two categories of responses under moderate stress, drought tolerance and drought resistance. The drought-tolerant strategy of a plant is to facilitate transpiration in order to increase the evaporative cooling effect, which also leads to a decrease in the photosynthetic carbon assimilation rate. In the drought-avoidance strategy, the plant is more likely to close its stomata when leaf surface temperature $T_s$ is too high or specific humidity $q_a$ is too low,




in order to reduce transpiration and water loss. Under severe water stress, both strategies lead to a large decrease in stomatal conductance and thus transpiration.

### 2.2.4 Soil moisture

The last parameter tested for its influence on evapotranspiration is soil moisture. It is essential as it directly influences plant
transpiration $E_{tr}$ by modulating $R_s$ through plant water stress and soil evaporation $E_g$ through $hu$, which is a function of soil moisture through Eq. 5.

$$hu = \begin{cases} 0.5 \left[ 1 - cos \left( \frac{w_g}{w_{fc}} \pi \right) \right] & \text{if } w_g \leq w_{fc} \\ 1 & \text{if } w_g \geq w_{fc} \end{cases}$$  (5)

To allow comparison of soil moisture values across different soil textures, the Soil Wetness Index (SWI) is often used in LSM parameterizations (Noilhan and Planton, 1989; Best et al., 2011; European Centre for Medium-Range Weather Forecasts, 2015)
and is also used in this article. The SWI is defined by the Eq. (6).

$$SWI = \frac{w_g - w_{wilt}}{w_{fc} - w_{wilt}}$$  (6)

where $w_g$ is the volumetric soil moisture, $w_{wilt}$ is the volumetric soil moisture at the wilting point, and $w_{fc}$ is the volumetric soil moisture at field capacity, all in $m^3 \, m^{-3}$. A $SWI$ value of 0 means the soil is at wilting point, and a value of 1 means the soil is at field capacity. Very dry soils may have negative values, and wet soils may have values greater than 1, up to saturation.

In order to assess the behaviour of the modelled evapotranspiration in different configurations, the parameters discussed above are all tested with different fixed SWI values. These fixed values allow the effect of different levels of water stress on transpiration to be clearly identified. In addition to the fixed SWI runs, ISBA is also run in its standard mode, with an evolving soil moisture without irrigation (*NOIRR*), and with irrigation parameterizations. The first parameterization keeps the soil at field capacity and is called *IRR_FC*. *NOIRR* and *IRR_FC* are used in both coupled and offline simulations. Another type of
irrigation parameterization is used in the offline simulations since it is also commonly used in the literature (Lawston et al., 2015; Wu et al., 2018b; Liu et al., 2021). These parameterizations add 30 mm or 100 mm of water to the top soil layer when the SWI falls below the 0.5 threshold. They are called *IRR_THLD*.

### 2.3 Evapotranspiration with FAO56

Another common way of estimating crop evapotranspiration is the approach used by the United Nations Food and Agriculture
Organization of the united nations (FAO), based on a Penman-Monteith formulation and described in Allen et al. (1998). This equation is widely used for evapotranspiration estimates in agronomy and hydrology (Oudin, 2005; Gavilán et al., 2007; Lemaitre-Basset et al., 2022). It is defined as

$$ET_0 = \frac{0.408 \Delta (R_n - G) + \gamma \left( \frac{900}{T_{mean}} \right) V_{a,2} VPD}{\Delta + \gamma (1 + 0.34 V_{a,2})}$$  (7)



where $ET_0$ is the reference evapotranspiration $[mm\ day^{-1}]$, $V_{a,2}$ is the daily mean wind speed at 2 m a.g.l. $[m\ s^{-1}]$, $R_n$ is
the net surface radiation $[MJ\ m^{-2}\ day^{-1}]$, $G$ is the ground heat flux $[MJ\ m^{-2}\ day^{-1}]$, $T$ is the mean daily air temperature
at 2 m $[K]$, $VPD$ is the mean daily vapour pressure deficit $[kPa]$, $\Delta$ is the slope of the vapour pressure curve $[kPa\ K^{-1}]$,
and $\gamma$ is the psychrometric constant $[kPa\ K^{-1}]$. The numbers 0.408, 900 and 0.34 are factors used for unit conversion and to
account for the surface characteristics assumed by the equation. In particular this reference evapotranspiration was designed
to represent the evapotranspiration over a surface of green grass about 0.12 m high, actively growing and well irrigated (Allen
et al., 1998). Since air temperature, humidity and wind speed are used in the calculation of the reference evapotranspiration,
the atmospheric feedback also directly influences the evapotranspiration estimates calculated with (7). The influence of the
atmospheric feedback on these estimates is therefore also examined below.

Note that evapotranspiration values can be expressed in different units as detailed in Appendix A. In the following, the values
of evapotranspiration are given both in terms of heat flux in W m$^{-2}$ and in terms of vapour flux in mm h$^{-1}$.

## 2.4 Atmospheric data

To study the atmospheric feedback on evapotranspiration, it is necessary to have two atmospheric data sets, one with and
one without irrigation effects on the atmosphere. To obtain these two different atmospheric data sets, the surface–atmosphere
coupled model Meso-NH is used. Meso-NH includes ISBA as LSM and has been shown to be able to efficiently represent
the effect of irrigation on near-surface atmospheric conditions and in the ABL (Lunel et al., 2024a, b). For the purpose of this
study, Meso-NH is run during the two weeks of the LIAISE SOP, at a 2 km horizontal resolution, with and without irrigation
activated. The Meso-NH configuration used here is the same as in Lunel et al. (2024b) and is fully described in that article. The
lowest atmospheric vertical level of the model is at 2 m above ground level (a.g.l.). The runs with and without irrigation are
named *NOIRR* and *IRR_FC* respectively, as in Lunel et al. (2024b). The *NOIRR* run uses a configuration similar to that used
for the limited-area operational NWP system AROME (Seity et al., 2011), and the SAFRAN reanalysis (Vidal et al., 2010; Le
Moigne et al., 2020).

The *IRR_FC* run simply adds an irrigation parameterization that keeps the soil moisture of irrigated areas at a constant field
capacity throughout the simulation. This parameterization has been shown by Lunel et al. (2024a) and Lunel et al. (2024b) to
be particularly good for modelling land-atmosphere fluxes and atmospheric conditions over the irrigated areas in the LIAISE
domain for two case study days.

To evaluate the modelled impact of irrigation on the atmosphere during the SOP, the model outputs are compared with data
from two in situ stations. The validation of the modelled irrigation impact for the two weeks of the SOP is discussed in Sect. 3.
These two stations are the La Cendrosa alfalfa field (Canut, 2022) and the IRTA corn field Martínez-Villagrasa et al. (2022),
also described and discussed in Boone et al. (2024), located in the hamlet of La Cendrosa and at the Institut de Recerca i
Tecnologia Agralimentàries (IRTA) facility, respectively. Both locations are well within the irrigated area.

The La Cendrosa field station is installed over a flood-irrigated alfalfa field. Numerous instruments have been installed at
this site, but the present work uses only the temperature, humidity and wind sensors located at 2 m a.g.l.. Alfalfa is on average
30 cm high during the SOP, and thus the displacement height can be estimated to be about 21 cm (Otsuki et al., 1999). In





aerodynamic terms, the La Cendrosa values for temperature, humidity and wind are therefore assumed to be 1.8 m above the surface. The IRTA corn field station is located about 1 km from the urban area of Mollerussa, above a flood-irrigated maize field

within the IRTA research facility. The station is placed between two rows of maize, about 2 m apart. The soil between the two rows was almost bare, but elsewhere in the field the maize was well grown, with a mean canopy height of 2.3 m, corresponding to a displacement height of approximately 1.6 m (Jacobs and Van Boxel, 1988; Otsuki et al., 1999). The anemometer is located at 3.3 m a.g.l. and the thermohygrometer at 2.5 m a.g.l., corresponding to 1.7 m and 0.9 m above the displacement height respectively.

The heights of the instruments at the two sites, La Cendrosa and the IRTA corn field, were close to 2 m above the displacement height. The observations are therefore compared directly with the model output at 2 m a.g.l., which corresponds to an explicit atmospheric model level.

The current combination of the coupled Meso-NH model with the two in situ observations stations enables a detailed quantification of the influence of irrigation on the atmosphere in Sect. 3, before studying the atmospheric feedback on evapotran-

spiration in Sect. 4.

## 3   Influence of irrigation on the atmosphere near the surface

Before studying how the atmosphere influenced by irrigation affects evapotranspiration, i.e. the atmospheric feedback, the influence of irrigation on the atmosphere needs to be clearly assessed and quantified. To this end, the present article focuses on the entire duration of the LIAISE SOP, from 14 to 30 July. Since the near-surface atmospheric features that most influence

evapotranspiration are air temperature $T_a$, specific humidity $q_a$ and wind speed $V_a$, the effect of irrigation on these three variables is examined for the SOP below. These variables taken from the coupled simulations are the inputs that are then used to build the atmospheric forcings used in the offline simulations of Sect. 4.

It should also be noted that incoming shortwave radiation, the most influential driver of evapotranspiration (Mangan et al., 2023; González-armas et al., 2024), was found to be globally unaffected by irrigation in the coupled model for the SOP. This

is firstly because there was little cloudiness either in reality or in the model, and secondly because most of this cloudiness was due to synoptic scale perturbations (Udina et al., 2024). However, there are some small differences in the shortwave radiation values for some days between the simulations, and in order to neutralize this weak effect, the incident shortwave radiation is set to the same values in both atmospheric forcings. Note that the $CO_2$ level, which is another driver of evapotranspiration (González-armas et al., 2024), is the same in both coupled simulations and subsequently in both atmospheric forcings.

### 3.1   Impact on air temperature

Between 14 and 30 July 2021, the observed and modelled mean diurnal cycles of near-surface air temperature are shown in Fig. 1 for the locations of the IRTA corn field and La Cendrosa alfalfa field sites. The difference between the coupled simulations *IRR_FC* and *NOIRR* allows quantification of the mean cooling effect of irrigation, which is found to range between -1.5





and -3°C for the modelled 2 m air temperature, depending on the time of the day. The cooling effect of irrigation significantly
reduces the discrepancy between observations and the simulated results from the *IRR_FC* coupled run.

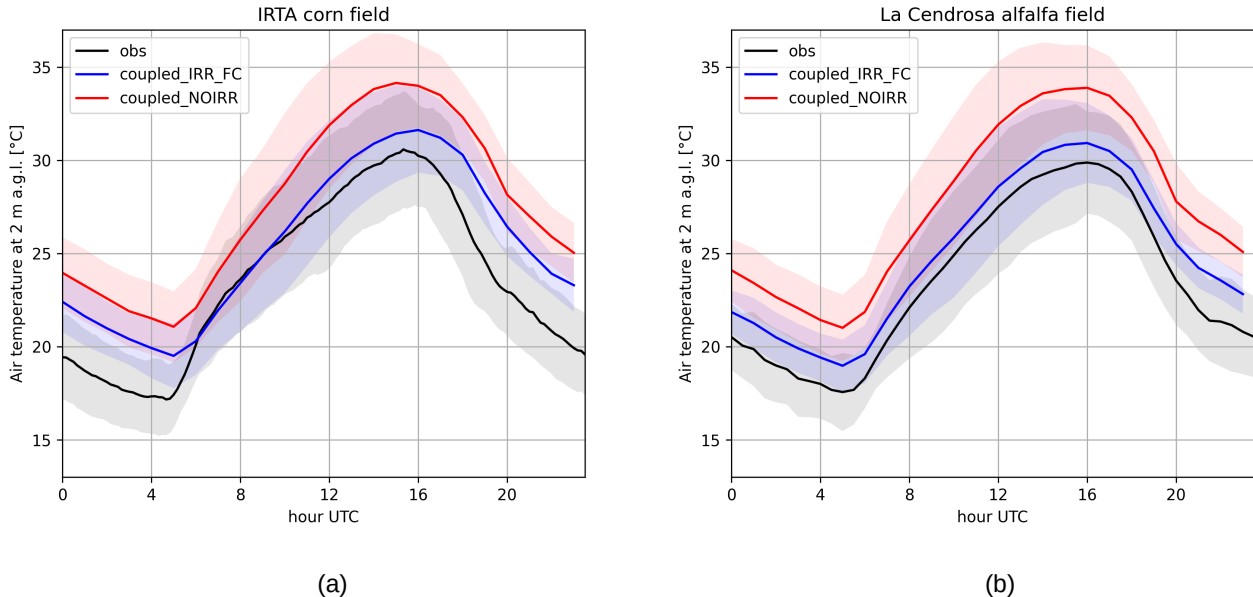

**Figure 1.** Mean modelled and observed 2-m air temperature diurnal cycle during the 15 days of the LIAISE SOP for the IRTA corn field in
(a) and for the La Cendrosa alfalfa field in (b). The red and blue lines correspond to the output of the surface–atmosphere coupled runs, with
and without irrigation, respectively. The black line represents the observed values. The shades of colour correspond to the standard deviation
of the data.

In the morning, between 06:00 and 09:00 UTC over the IRTA corn field (Fig. 1 (a)), the coupled simulation with irriga-
tion manages to accurately model the air temperature. During afternoons and nights, this coupled simulation moves away
from the observations. This behaviour suggests that the model has inherent biases that vary over the course of the day, and
that the overestimation of air temperature by the coupled simulation *IRR_FC* during the afternoon and night is not due to a
misrepresentation of irrigation effects, as it is well represented in the morning.

Over the alfalfa field of La Cendrosa (Fig. 1 (b)) the modelled cooling effect is slightly more important than over the IRTA
corn field. It is about -3°C throughout the day. The remaining overestimation of air temperature by the model *IRR_FC* compared
to observation is relatively constant throughout the day. It is about 1°C during the day and about 1.5°C during the night. For the
La Cendrosa alfalfa field, as for the IRTA corn field, the remaining overestimation of the night air temperature by the *IRR_FC*
simulation is most likely due to the difficulty of the model to represent the stable conditions of the night (Bravo et al., 2008;
Holtslag et al., 2013).

The performance scores of the coupled model runs with and without irrigation are shown in Tables 3 and 4, for the IRTA
corn field and the La Cendrosa alfalfa field, respectively. The scores are given for daytime, as it is mainly the daytime values





that influence evapotranspiration, and also for the whole period to give a general view of the model behaviour. The average
cooling effect of irrigation over the whole period is -2.21°C and -2.65°C for the IRTA corn field and the La Cendrosa alfalfa
field, respectively, and -2.40°C and -2.87°C for the daytime only. The remaining temperature bias found between the *IRR_FC*
coupled simulation and the observations depends on the day. Lunel et al. (2024a) showed that for 21 and 22 July 2021, adding
irrigation to the coupled model causes a reduction in most of this bias, but for some other periods the bias is higher. This may
be due to the fact that even if the activation of irrigation improves the model representation of winds, some wind regimes may
be missed. As the topography of the region is complex, it is expected that the models do not capture all of the mesoscale winds
(Jiménez et al., 2024). In addition, the coupled model has inherent biases, which may be related to the modelling of stable
atmospheric conditions, the composite approach used in SURFEX, biases in meteorological variables derived from analyses
used at the boundaries of the simulation domain, or potential misrepresentation of the land surface in the vicinity of the Urgell
area. In particular, the vegetation in the model around the irrigated area may have too shallow roots as discussed by Canal
et al. (2014); Shrestha et al. (2018), leading to an overestimation of plant water stress and ultimately to warmer air. The precise
characterization of the internal model biases in this region would require extensive further work, which is considered beyond
the scope of the current study. Lunel et al. (2024b) and Lunel et al. (2024a) showed that irrigation parameterizations that
maintain SWI values within irrigated areas close to field capacity perform very well. Therefore, in the context of this work, it
is assumed that the remaining overestimation of air temperature is not due to a misrepresentation of the irrigation effect, but to
other shortcomings of the model.

## 3.2   Impact on specific humidity

The mean effect of irrigation on near-surface specific humidity is shown in Fig. 2. In the *IRR_FC* coupled simulation, the mean
specific humidity over the IRTA corn field is overestimated throughout the day (Fig. 2 (a)). The overestimation is specifically
important at 08:00 UTC, reaching up to 2 g kg$^{-2}$. This overestimation of humidity in the morning is due to the lack of vertical
mixing in the lower ABL, as already discussed in Lunel et al. (2024a). During the afternoon, the simulation *IRR_FC* reduces
the discrepancy with the observation, although it still overestimates the absolute value. The difficulty of the simulation in
modelling the specific humidity at the IRTA corn field could be due to several shortcomings of the coupled model, such as
the misrepresentation of the neighbouring town of Mollerussa or the overrepresentation of irrigated areas in the immediate
vicinity of the IRTA corn field. In any case, the difference between the simulations *NOIRR* and *IRR_FC* allow to estimate the
humidifying effect at the IRTA facility, and is found to be +2.31 g kg$^{-1}$ during daytime, and +1.50 g kg$^{-1}$ globally (Table 3).

At the La Cendrosa alfalfa field the specific humidity during the day is well modelled by the simulation *IRR_FC* as shown
in Fig. 2 (b). At night, the specific humidity is slightly underestimated, probably due to the difficulty of the model in modelling
stable nighttime conditions as discussed above. At all times the simulation *IRR_FC* improves the underestimated humidity of
the simulation *NOIRR*. The humidifying effect at La Cendrosa is found to be +2.68 g kg$^{-1}$ during daytime, and +1.78 g kg$^{-1}$
globally (Table 4).





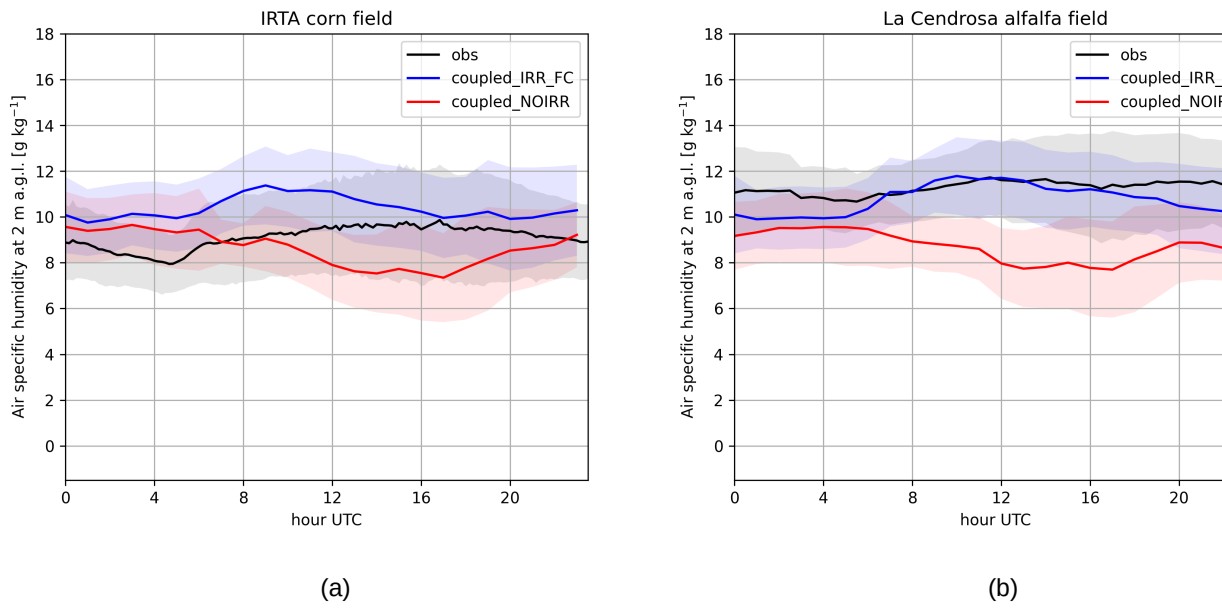

**Figure 2.** As in Fig. 1 but for the 2-m specific humidity.

## 3.3 Impact on wind speed

Many studies have shown that irrigation can slow the wind near the surface (Segal et al., 1989; Sorooshian et al., 2011; Sridhar, 2013; Wu et al., 2018a). This is also the case in the Urgell region, as shown in Lunel et al. (2024a) for the 21 and 22 July, and in Fig. 3 for the whole SOP at the two observation sites. The mean reduction in wind speed due to the irrigation parameterization

is -0.60 and -0.68 m s$^{-1}$ for the corn and alfalfa field, respectively, during the SOP (Tables 3 and 4). During the day only, the reduction in wind speed due to irrigation is -0.86 and -0.94 m s$^{-1}$, respectively. This reduction in wind speed is mainly due to the reduced momentum flux between the free troposphere and the surface (Sridhar, 2013; Wu et al., 2018a). However irrigation can also influence the dynamics of mesoscale winds which are modulated by the complex topography of the Urgell region (Lunel et al., 2024a, b). In particular, the Marinada is the mesoscale wind that blows strongest during the SOP. It corresponds

to the peak wind speed found at 17:00 and 19:00 UTC for the *NOIRR* and *IRR_FC* coupled simulations in Fig. 3 (a) and (b), respectively. In particular, both figures show that the peak wind speed is delayed by irrigation and that the maximum wind speed is significantly reduced by up to 1.4 m s$^{-1}$, thus confirming the findings of Lunel et al. (2024b) over the longer period of the LIAISE SOP.

The reduction in wind speed due to irrigation allows a more accurate representation of the wind speed in the coupled model

compared to observations. Fig. 3 (b) shows that the irrigated coupled simulation *IRR_FC* performs very well in representing the wind speed at the La Cendrosa alfalfa field. The irrigation parameterization also improves the representation of wind speed over the IRTA corn field, but some biases remain. Figure 3 (a) shows a relatively poor agreement between the irrigated simulation





and the observations for the IRTA corn field, especially for the evening and night. Since the modelled wind speed behaviour is very similar in the IRTA corn field as in La Cendrosa, the reason for the poor agreement is more related to the observed wind

speed pattern of the IRTA corn field, which is different from that of La Cendrosa mainly because it does not show faster winds between 16:00 and 20:00 UTC, which should correspond to the Marinada. A possible explanation for this is the organization of crop canopies into rows in and around the corn field. As shown by Ulmer et al. (2023), the angle between the rows and the wind can influence the wind speed up to two times the canopy height. This is the case for the corn field where the wind speed sensor is located at 3.6 m a.g.l., in a 2-m wide linear corridor between two rows of corn, with a surrounding canopy at 2.3 m.

The rows of corn and most of the surrounding crops are aligned on a southwest-northeast axis. This is almost parallel to the dominant west-southwest wind that blows in the Urgell region, and therefore the crops offer less resistance to the wind coming from this direction. Conversely, the Marinada is a southeasterly wind that is almost perpendicular to the rows, which means that the wind measured may be weaker. In other words, the west-southwest wind sees a relatively lower roughness length than the southeast wind. However, the effect found by Ulmer et al. (2023) is relatively small, about 0.1 to 0.2 m s$^{-1}$ at 1.7 times

the canopy height. Although the field studied in Ulmer et al. (2023) is not the same, it is delicate to assume that all the bias between the observations and the irrigated simulation comes from the observational data. In conclusion, the reason for the absence of the Marinada signal in the wind speed measurements over the corn field would require other types of observations and an in-depth investigation, which is not undertaken in the present work.

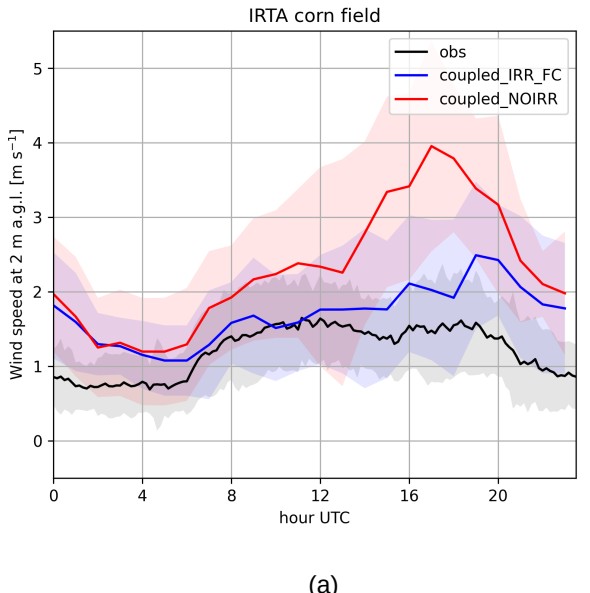
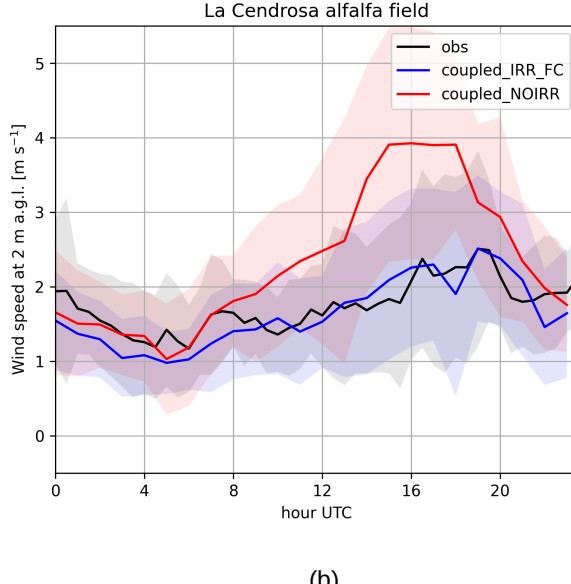

(a)                                                    (b)

**Figure 3.** As in Fig. 1 but for the 2-m wind speed.





**Table 3.** Performance scores for the two coupled simulations *NOIRR* and *IRR_FC* for the SOP at the IRTA corn field. The subscripts *global* and *daytime* indicate the periods on which the score has been calculated, respectively, over the whole period and during the day, i.e. between 05:00 and 19:00 UTC.

|  | $T_a$ [°C] | | $q_a$ [g kg$^{-1}$] | | $V_a$ [m s$^{-1}$] | |
|---|---|---|---|---|---|---|
|  | IRR_FC | NOIRR | IRR_FC | NOIRR | IRR_FC | NOIRR |
| Bias$_{global}$ | 1.01 | 3.22 | 1.23 | -0.27 | 0.54 | 1.14 |
| RMSE$_{global}$ | 1.97 | 3.57 | 1.75 | 1.77 | 0.92 | 1.53 |
| Bias$_{daytime}$ | 1.18 | 3.58 | 1.32 | -0.99 | 0.31 | 1.17 |
| RMSE$_{daytime}$ | 1.63 | 3.76 | 1.44 | 1.47 | 0.38 | 1.35 |

**Table 4.** As for Table 3, but for the La Cendrosa alfalfa field.

|  | $T_a$ [°C] | | $q_a$ [g kg$^{-1}$] | | $V_a$ [m s$^{-1}$] | |
|---|---|---|---|---|---|---|
|  | IRR_FC | NOIRR | IRR_FC | NOIRR | IRR_FC | NOIRR |
| Bias$_{global}$ | 1.2 | 3.85 | -0.46 | -2.24 | -0.08 | 0.60 |
| RMSE$_{global}$ | 1.66 | 4.03 | 1.26 | 2.72 | 0.87 | 1.40 |
| Bias$_{daytime}$ | 1.14 | 4.01 | -0.19 | -2.87 | -0.06 | 0.88 |
| RMSE$_{daytime}$ | 1.16 | 4.03 | 0.40 | 3.00 | 0.23 | 1.16 |

This section confirms the strong influence of irrigation on air temperature, humidity and wind speed, and quantifies these effects during the LIAISE SOP with surface–atmosphere coupled simulations. Although the coupled model proves to be able to represent well this influence, it is also shown that biases remain between the model with irrigation and the observations. This could be due to problems inherent in the model or also to some measurement uncertainties. For the remainder of this study, it is assumed that among the various potential model shortcomings, the remaining biases are not so much due to misrepresentation of irrigation effects, but rather mostly to other model shortcomings.

## 4   Evaluation of the atmospheric feedback on evapotranspiration

The previous section demonstrated the significant effect of irrigation on the near-surface atmosphere: the atmosphere becomes cooler, more humid, and less windy. These effects, in turn, affect the surface and, in particular, surface evapotranspiration. This is the atmospheric feedback. This feedback can be consistently represented in coupled surface–atmosphere models, as both the surface and atmosphere models interact at each time step. However uncoupled (i.e. offline) LSMs do not represent the atmospheric feedback. Since coupled models are computationally expensive to run, other research areas such as hydrology and agronomy also use LSMs, but without coupling the surface to the atmosphere. These offline simulations can either be run for a single point (or parcel), or they can be made over a 2D domain. In the former case, the LSM can be forced by observations if available. In either case, the atmospheric conditions can originate from atmospheric model data (hindcast or a





forecast) taken directly from coupled simulation outputs. Typically, the atmospheric forcing data used come from reanalysis
(an optimal combination between model and observations) for past or near-real time weather, or GCM for future weather.
However, irrigation is rarely considered in meteorological and climatological simulations (Mcdermid et al., 2023), and the
reanalysis and GCM near-surface atmospheric state variables are very often too dry, too warm, and too windy over irrigated
areas in summer (Tuinenburg and de Vries, 2017; Qian et al., 2020). The offline LSMs runs can also represent the soil moisture
modifications due to irrigation, thus increasing crop evapotranspiration, but without having the effect of irrigation on the
atmosphere represented in the atmospheric forcing, i.e. without accounting for the atmospheric feedback. The purpose of this
section is to assess how this warm, dry and windy bias affects the evapotranspiration modelled by the offline ISBA LSM with
a particular focus over irrigated areas.

A primarily model-based methodology is used to study the influence of atmospheric feedback on evapotranspiration. Point
scale offline simulations are performed using atmospheric forcings that either include or not the effect of irrigation on the
atmosphere. These atmospheric forcings are generated directly from the output of the coupled simulations presented in Sect. 3,
namely *NOIRR* and *IRR_FC*, and are hereafter named *atmo_NOIRR* and *atmo_IRR_FC* to avoid confusion with offline LSM
simulations using these irrigation parameterizations. The atmospheric forcings generated are based on the atmospheric condi-
tions modelled over the hamlet of La Cendrosa and the IRTA facility. It should be noted that from here on out, references to
La Cendrosa and the IRTA facility only indicate different atmospheric conditions and, in particular, different irrigation effects
on the atmosphere, but do not imply a specific underlying land surface, as it was previously the case with the corn and alfalfa
field. Here the atmospheric forcing *atmo_NOIRR* plays the role of the standard reanalysis or GCM data commonly used in
offline LSMs. The results of these offline simulations are then compared on the basis of the presence or absence of irrigation
in the atmospheric forcing. By comparing only the model results, it is assumed that the biases inherent in the model are largely
cancelled out when calculating the difference between the results of the two configurations. The offline models are chosen to be
run only at the point scale, rather than in a 2D spatial domain, to allow for a detailed interpretation of how evapotranspiration
is influenced by the atmospheric conditions that have been finely characterized in Sect. 3.

Sect. 4.1 presents results using a fixed configuration of offline ISBA. This allows the investigation of the processes that
explain the behaviour of ISBA by focusing on specific days of the SOP. Sect. 4.2 then generalizes the results by combining
different possible offline model configurations. This enables a quantification of the evapotranspiration overestimation due to
the *atmo_NOIRR* atmospheric forcing for the whole SOP.

## 4.1 Processes at play

This section aims to better understand the ISBA processes that regulate evapotranspiration under different atmospheric condi-
tions. To focus on the effects of atmospheric conditions, a fixed surface configuration is maintained which includes a single
Plant Functional Type (PFT) representing a drought-tolerant crop, a composite approach for the vegetation representation, a
stomatal conductance modelled with ISBA-A-$g_s$. The atmospheric forcing used corresponds to the near-surface atmospheric
conditions found over the IRTA facility at 2 m a.g.l. in the coupled model runs. The rest of the offline ISBA configuration
corresponds to the values presented in Sect. 2. Note that even though the near-surface atmospheric conditions are those of the





IRTA facility, the underlying surface of the offline run is not intended to be representative of the actual IRTA corn field. This allows the effect of atmospheric feedback on evapotranspiration to be isolated from the effects of surface characteristics on
evapotranspiration.

### 4.1.1 Without water stress

When a crop is well irrigated, most of the evapotranspiration is driven first by incoming (downward) shortwave radiation and secondly by air temperature, humidity and wind. On 15 July 2021, no clouds were modelled in Meso-NH, and therefore the incoming shortwave radiation was essentially the same in the two atmospheric forcings *atmo_IRR_FC* and *atmo_NOIRR*. Thus,
for the two offline ISBA runs, the factors for the differences in evapotranspiration are found in air temperature, humidity and wind.

Figure 4 shows the different atmospheric conditions from the two atmospheric forcing dataset for 15 July 2021 at the IRTA facility, and the resulting evapotranspiration modelled by the offline ISBA. The point-scale offline ISBA simulation has a $SWI$=1.0. The cumulative evapotranspiration modelled with the *atmo_NOIRR* atmospheric forcing, i.e. with a warm, dry,
and windy atmosphere, is 21% higher than that produced with the *atmo_IRR_FC* atmospheric forcing. In other words, all other things being equal, neglecting the irrigation effect in the atmospheric forcing leads to a 21% overestimation of evapo-transpiration for 15 July 2021. This corresponds to an overestimation of the daily mean latent heat flux by 36 W m$^{-2}$, or an overestimation of the cumulative daily evapotranspiration by 1.37 mm d$^{-1}$.

### 4.1.2 Under water-stress conditions - role of stomatal closure

In irrigated regions, not all crops are necessarily irrigated. These rainfed crops are therefore exposed to atmospheric conditions influenced by irrigation without being irrigated themselves. In this case, the plant may be under water stress and may close its stomata to limit water loss. As stomatal closure is controlled not only by soil moisture but also by temperature, specific humidity and wind speed, the atmospheric feedback also affects stomatal closure, and eventually transpiration. Note that stomatal closure is also controlled by shortwave radiation and $CO_2$ concentration, but these effects are neutralized in the
present study as discussed previously.

Figure 5 shows the transpiration modelled by offline ISBA for two different days, for a $SWI$ value of 0.2, and with the two different atmospheric forcings *atmo_NOIRR* and *atmo_IRR_FC* at the IRTA facility. On 18 July (Fig. 5 (a)), the temperature is higher than for the 15 July conditions shown in Fig. 4, and the model represents stomatal closure for the simulation with the *atmo_NOIRR* atmospheric forcing. The stomatal closure is characterized in the time series of transpiration vapour flux by
a sudden drop in the flux value found during the day. On 18 July, the stomata close between 12:00 and 15:00 UTC, when the air temperature is highest. The asymmetry of stomatal closure around midday confirms the strong effect of air temperature. Conversely, no stomatal closure is modelled for the simulation using the *atmo_IRR_FC* atmospheric forcing. In this case, the atmospheric feedback, i.e. the cooling, humidifying effect and the weakening of the wind, prevents stomatal closure. Counter-intuitively, the cumulative transpiration for this day is higher for the simulation with the *atmo_IRR_FC* atmospheric forcing,
i.e. with cooler, more humid and less windy atmospheric conditions. The simulation with *atmo_NOIRR* underestimates the





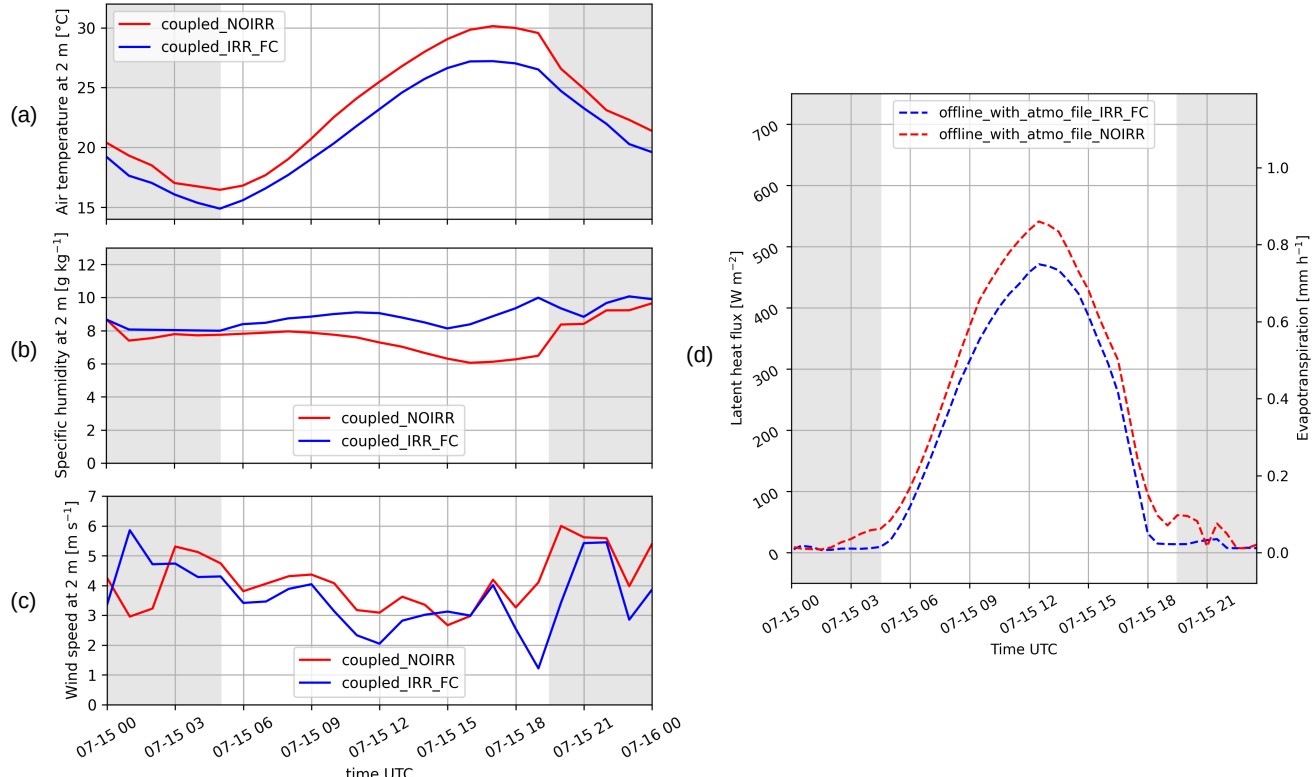

**Figure 4.** Atmospheric conditions and evapotranspiration modelled by ISBA with two different atmospheric forcings, for 15 July 2021 at the IRTA facility. On the left are the air temperature (a), specific humidity (b) and wind speed (c), and on the right the resulting evapotranspiration (d). Both simulations have soil moisture set to field capacity, represent a drought-tolerant PFT, use a composite approach for the vegetation canopy, and model stomatal conductance with the ISBA-A-$g_s$ parameterization.

transpiration rate by 29% compared to the simulation with *atmo_IRR_FC*. This corresponds to an underestimation of the daily mean transpiration latent heat flux by 30 W m$^{-2}$, or an underestimation of the cumulative daily transpiration by 1.14 mm d$^{-1}$.

On a hotter day like 22 July 2021, both simulations show stomatal closure (Fig. 5 (b)). However, the atmospheric feedback still allows stomata to close later in the morning. The simulation with the *atmo_IRR_FC* atmospheric forcing has stomatal clo-
sure between 12:00 and 15:00 UTC, while the *atmo_NOIRR* forcing results in stomata closure between 10:00 and 14:30 UTC. Somewhat unexpectedly, stomata reopen earlier in the case without atmospheric feedback. This is due to the earlier arrival of the Marinada in the *atmo_NOIRR* atmospheric forcing, as shown by Lunel et al. (2024b). This behaviour is therefore specific to the region of the present case study and cannot be extended to other regions of the world. Without the Marinada delay, the atmospheric feedback should have led to an earlier reopening of the stomata. The daily transpiration difference is again
negative, meaning that the atmospheric feedback led to higher transpiration rates. The offline simulation with *atmo_NOIRR*



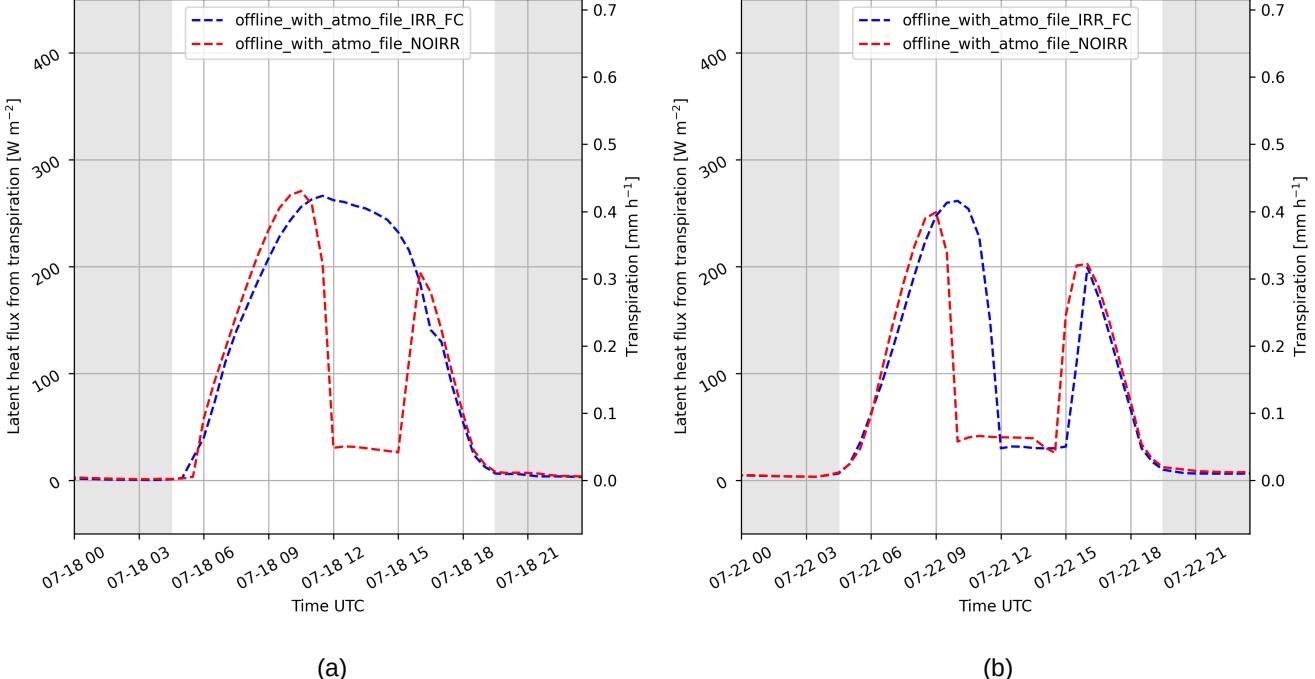

**Figure 5.** Transpiration modelled by ISBA for 18 (a) and 22 (b) July 2021 over a hypothetical dry parcel ($SWI$=0.2) located at the IRTA facility. The PFT modelled here corresponds to a drought-tolerant crop, with vegetation modelled with a composite approach and stomatal conductance modelled by ISBA-A-$g_s$.

atmospheric forcing, i.e. without atmospheric feedback, underestimates the transpiration rate by 10%. This corresponds to an underestimation of the daily mean transpiration latent heat flux by 7 W m$^{-2}$, or an underestimation of the cumulative daily transpiration by 0.28 mm d$^{-1}$.

However, transpiration is only part of evapotranspiration, the other consisting in evaporation from the bare ground (there is no water intercepted on the leaves). The total evapotranspiration is shown in Fig. 6. Stomatal closure can still be seen in the daily evolution of total evapotranspiration, however, the decrease in total evapotranspiration is not as significant as for transpiration only.

On 18 July at 13:00 UTC, the instantaneous decrease due to stomatal closure for transpiration only is about 220 W m$^{-2}$, while the decrease for evapotranspiration is about 100 W m$^{-2}$. The difference in the magnitude of the decrease between transpiration and evapotranspiration is due to a compensating effect of bare soil evaporation. In the composite approach, as vegetation transpiration decreases, more heat is available to the composite layer, and the soil responds quickly by increasing its temperature. In the separate canopy approach of MEB, as transpiration decreases, the canopy converts more radiative energy



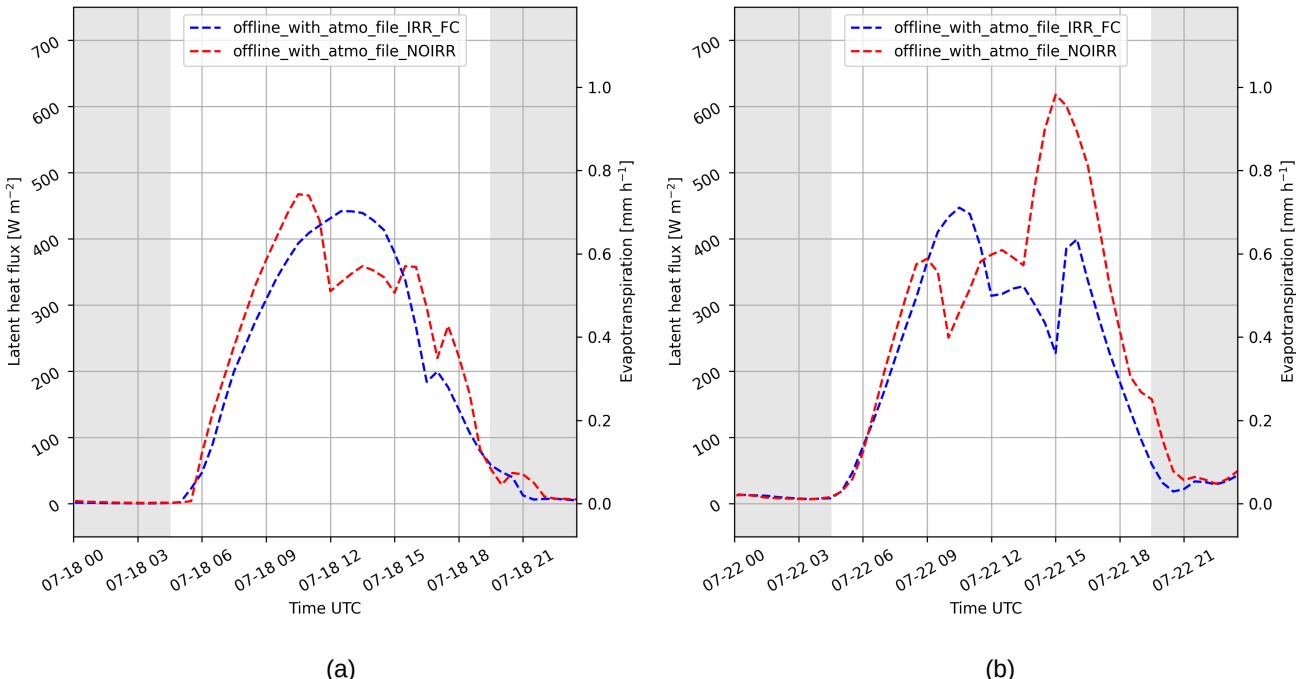

(a)                 (b)

**Figure 6.** As in Fig. 5, but for evapotranspiration.

into thermal energy, increasing the canopy air temperature and also the top soil temperature. Under either approach, the soil warms as transpiration decreases and evaporation from the bare soil increases in response.

Another interesting effect due to evaporation can be seen in Fig. 6. Soil evaporation is also more sensitive to wind speed than transpiration, and evaporation peaks when the sustained wind speed of the Marinada arrives. Combining Eq. 1 and 3, it can be observed that the ISBA LSM formulates evaporation as proportional to wind speed $V_a$, while transpiration is also modulated by stomatal resistance $R_{s,leaf}$. This strong dependence on the wind speed allows the modelled evapotranspiration to reach up to 615 W m$^{-2}$ (0.98 mm h$^{-1}$) on 22 July at 15:00 UTC when the Marinada arrives. In this case, it is important to

consider the role of the irrigation-induced decrease in wind speed. The excessively windy atmospheric forcing *atmo_NOIRR* leads to an overestimation of the mean evaporation up to 63% and 47% for 18 and 22 July, respectively (Appendix A1 (a, b)). These values correspond to an overestimation of the daily mean evaporation latent heat fluxes by 41 and 47 W m$^{-2}$, or an overestimation of the cumulative daily evaporation by 1.54 and 1.80 mm d$^{-1}$. By aggregating the role of evaporation and transpiration, for the days shown in Fig. 6, it can be found that the *atmo_NOIRR* atmospheric forcing leads to an overestimation

of evapotranspiration by about 6% and 23% for 18 and 22 July, respectively, corresponding to an overestimation of the daily mean latent heat fluxes by 10 and 39 W m$^{-2}$, or an overestimation of the cumulative daily evapotranspiration by 0.40 and 1.52 mm d$^{-1}$.





It should be noted that these results are obtained using the composite approach to vegetation representation, which is known
to have limitations with respect to bare ground evaporation (Napoly, 2016). By using the MEB option, the wind speed seen
by the ground surface can be modelled more realistically and is significantly lower than that seen by the canopy. The MEB
approach also takes into account the sheltering effect of the canopy and the temperature of the top layer of soil is independent
of the canopy temperature. This more realistic approach means that ISBA does not compensate for reduced transpiration by
increasing evaporation as much as in the composite approach. The overestimation of evaporation then decreases, becoming
negligible on 18 July, but remaining on 22 July. The mean evaporation is overestimated by 0.5% and 17% for 18 and 22 July,
respectively (Appendix A1 (c, d)). These values correspond to an overestimation of the daily mean evaporation latent heat
fluxes by 1 and 11 W m$^{-2}$, or an overestimation of the cumulative daily evaporation by 0.01 and 0.43 mm d$^{-1}$. With the
separate canopy approach of MEB, the behaviour of total evapotranspiration is closer to that of transpiration. On 18 July the
evapotranspiration is underestimated by 16% (24 W m$^{-2}$ or 0.92 mm d$^{-1}$), whereas on 22 July it is overestimated by only 5%
(6 W m$^{-2}$ or 0.24 mm d$^{-1}$).

Note that the atmospheric feedback on evaporation and the compensation effect must be interpreted with caution. Evapora-
tion from bare soil is strongly dependent on the top soil humidity $hu$. Although moisture can be evaporated from the soil for
soil moisture below the wilting point, i.e. for $SWI < 0$, the rate of evaporation will be low for such dry soils. Note that the
model configuration used in Fig. 6 is conceptual and keeps $SWI$ at a value of 0.2, which allows the top soil layer to be moist
enough to provide a significant evaporation rate throughout the simulation. In reality, or in a more realistic flood irrigation
parameterization, the top soil layer would dry out and evaporation could not compensate as much for the decrease in transpira-
tion. Although the magnitude of the evapotranspiration compensation effect is uncertain, it is important to consider this process
when interpreting the evapotranspiration outputs of LSMs in different configurations, as is done in the next section.

## 4.2  General quantification

The previous sections have shown that for equal incoming shortwave radiation, the atmospheric feedback is particularly impor-
tant to consider in offline ISBA simulations. For well-irrigated plants, the atmospheric feedback is shown to reduce modelled
transpiration and evaporation. For water-stressed plants, however, more complex processes are involved. The atmospheric feed-
back has a clear influence on the timing of stomatal closure, which can compensate for the decrease in transpiration or even
lead to the opposite effect, i.e. an increase in transpiration. It has also been shown that transpiration and soil evaporation are
not separate processes, but interact with each other through surface temperature. More specifically, a decrease in transpiration
leads to an increase in bare soil evaporation in the ISBA LSM.

The previous section explored and quantified these processes for specific days and for a few ISBA configurations. A more
systematic assessment of the effect of atmospheric feedback on vapour fluxes is still needed. For this purpose, the overestima-
tion of evapotranspiration due to the *atmo_NOIRR* atmospheric forcing is quantified for the different offline ISBA configura-
tions presented in Table 2. The influence of the different parameter combinations is also discussed in the light of the previously
presented processes.



**Figure 7.** Transpiration (left panel) and evaporation (right panel) mean absolute overestimation due to the *atmo_NOIRR* atmospheric forcing as a function of root zone SWI, at the two locations of the IRTA facility (a & b) and La Cendrosa (c & d). Dro.-tol., dro.-avo., and comp. stand for drought-tolerant, drought-avoidant, and composite approach, respectively. The purple and green lines correspond to the drought-avoidant and drought-tolerant strategies, respectively. The dotted and dashed lines correspond to the ISBA-A-$g_s$ and Jarvis photosynthesis strategies, respectively. The round and diamond markers correspond to the MEB and composite approach strategies, respectively.





Figure 7 shows the overestimation of transpiration and evaporation rates modelled by ISBA for different soil moisture values, different ISBA configurations (listed in Table 2) and for two different locations in the irrigated area of LIAISE. For transpiration, the absolute value of the overestimation is highly dependent on the soil moisture and the stomatal conductance scheme (Fig. 7 (a) and (c)). With Jarvis, the mean absolute overestimation of transpiration due to the lack of atmospheric feedback ranges between 28 and 56 W m$^{-2}$ (1.07 and 2.14 mm d$^{-1}$) for wet soil where $SWI \geq 1$. In relative terms, these values correspond to a 19% to 35% overestimation of the mean transpiration vapour flux. For dry soil ($SWI = 0.1$) the overestimation ranges from 1 to 21 W m$^{-2}$ (0.04 to 0.80 mm d$^{-1}$), corresponding to a relative overestimation of 4 to 21%. No value is shown for $SWI = 0$ as the modelled transpiration is zero. For these drier soils and with Jarvis, the decrease in the absolute value of the overestimation is due to the overall decrease in evapotranspiration, but also partly to the compensating effect of stomatal closure as presented in the previous section. Stomatal closure/opening is regulated by stomatal conductance and the associated scheme. Fig. 7 (a) and (c) show that the Jarvis scheme leads to higher overestimation of transpiration than the ISBA-A-$g_s$ scheme. In fact, the Jarvis scheme does not model rapid stomatal closure. Instead, it gradually reduces stomatal conductance during the day and leads to a weak compensation effect with respect to the timing of stomatal closure (not shown).

In contrast, when the ISBA-A-$g_s$ scheme is used, stomatal closure compensation is more significant and greatly reduces the overestimation of transpiration for all soil moisture compared to the Jarvis scheme. For ISBA-A-$g_s$ and wet soils, the lack of atmospheric feedback can lead to either an overestimation of 14 W m$^{-2}$ (0.54 mm d$^{-1}$) or an underestimation of -12 W m$^{-2}$ (-0.46 mm d$^{-1}$). This can be explained by the fact that the two weeks modelled for the present study are hot and dry in midsummer and therefore the modelled plants often close their stomata when the *atmo_NOIRR* atmospheric forcing is used, i.e. when it does not include the atmospheric feedback. However, the *atmo_IRR_FC* atmospheric forcing allows for more days without stomatal closure to be modelled. On these days the atmospheric feedback actually allows a higher cumulative transpiration rate, as shown in Fig. 5. In the present case, the ISBA-A-$g_s$ scheme used with hot and dry atmospheric conditions leads to a low or negligible overestimation of transpiration.

The response of evaporation to ISBA configuration and soil moisture is different from that of transpiration. Figure 7 (b) and (d) show the mean absolute overestimation of evaporation as a function of soil moisture. The canopy representation is the main factor to consider in this case, with a higher overestimation when the composite approach is used. With the composite approach, the absolute overestimation values range from 26 to 63 W m$^{-2}$ (0.99 to 2.40 mm d$^{-1}$) depending on the combination of stomatal conductance scheme, drought response and location. The reason for this is the dependence of evaporation on the temperature of the top soil layer. As the top soil layer is a composite layer, its temperature depends on the cooling caused by transpiration. The lower the transpiration, the higher the top soil temperature and ultimately the higher the evaporation, i.e. evaporation compensates, to a certain degree, for the lack of transpiration. This dependence on transpiration is confirmed by the fact that the absolute differences in evaporation rate do not vary with the stomatal conductance scheme for $SWI = 0$, i.e. when transpiration is zero. It can also be observed that the highest values of evaporation overestimation are found for ISBA-A-$g_s$, i.e. for the stomatal conductance scheme that models more stomatal closure and overestimates transpiration less.

With MEB, bare soil evaporation is more physical. It has independent temperatures for the canopy and the top soil layer and takes into account the shading effect of the canopy on the soil. The top soil temperature is not influenced as quickly by





stomatal closure as in the composite approach, and therefore the evaporation overestimation is globally similar for all soil moisture and ISBA configurations, with values ranging from 8 to 18 W m$^{-2}$ (0.31 to 0.69 mm d$^{-1}$). In relative terms, these values correspond to a 12% to 38% overestimation of the mean soil evaporation vapour flux.

In summary, the overestimation of transpiration is highly dependent on the soil moisture and stomatal conductance scheme, while the overestimation of evaporation is more dependent on the canopy scheme. For the composite approach, the higher the overestimation of transpiration in Fig. 7 (a) and (c), the lower the overestimation of evaporation in Fig. 7 (b) and (d).

Figure 8 shows the distribution of the total relative overestimations of evapotranspiration, for the two weeks modelled, for different ISBA configurations. These configurations differ in the parameterizations used (see Table 2) and in the atmospheric effects of irrigation (from La Cendrosa and the IRTA facility). The detailed relative overestimation for each configuration is shown in Appendix A2. The results are presented separately for different soil moisture categories in offline ISBA: *Non-irrigated* and *Irrigated*. The *Non-irrigated* column corresponds to the *NOIRR* case and the *Irrigated* column corresponds to the *IRR_THLD* and *IRR_FC* parameterizations, described in Sect. 2.2.4.

When the soil is not irrigated in the offline simulation, the stomatal closure process plays a key role, and the use of *atmo_NOIRR* leads to weak or negligible average underestimation of evapotranspiration, between -10 and 2%, as discussed in Sect. 4.1.2. Moreover, this small relative underestimation corresponds to small absolute values. This means that a field that is not irrigated, but is subject to atmospheric feedback from surrounding irrigated fields, will not reduce its mean evapotranspiration very significantly.

In contrast, the irrigated field is quite sensitive to the atmospheric feedback. In this case, the use of the atmospheric forcing *atmo_NOIRR*, which does not include the influence of irrigation on the atmosphere, leads to an overestimation of evapotranspiration of between 4% and 35%. For the irrigated offline simulations, the low values of evapotranspiration overestimation (< 20%) are all obtained for ISBA configurations using the stomatal conductance scheme ISBA-A-$g_s$, since this scheme allows more stomatal compensation effect (Appendix A2). A LSM using the Jarvis stomatal conductance scheme is more prone to overestimate evapotranspiration over irrigated areas.

Evapotranspiration can also be calculated without LSM, for example using a formula such as FAO-56 from Eq. 7. This formula implies a wet soil and crops without water stress, thus conceptually representing a land surface similar to that modelled with the irrigated offline ISBA simulations. By entering values for $V_{a,2}$, $T$ and $VPD$ either taken from *atmo_NOIRR* or *atmo_IRR_FC*, the overestimation due to the lack of atmospheric feedback in the input data can also be evaluated. The overestimation is found to be similar with the FAO-56 formula as with the offline ISBA simulations, i.e. between 20% and 24%. The average overestimation for all cases including irrigation (i.e. *Irrigated* and *FAO-56*) is 25%.

## 5 Discussion and conclusions

This article examines the importance of accounting for irrigation in atmospheric forcings that are subsequently used for hydrological or agronomic purposes over irrigated areas. The influence of irrigation on near-surface meteorological conditions, i.e. air temperature, humidity and wind, is shown to be important, and it is quantified for the two weeks of the LIAISE campaign



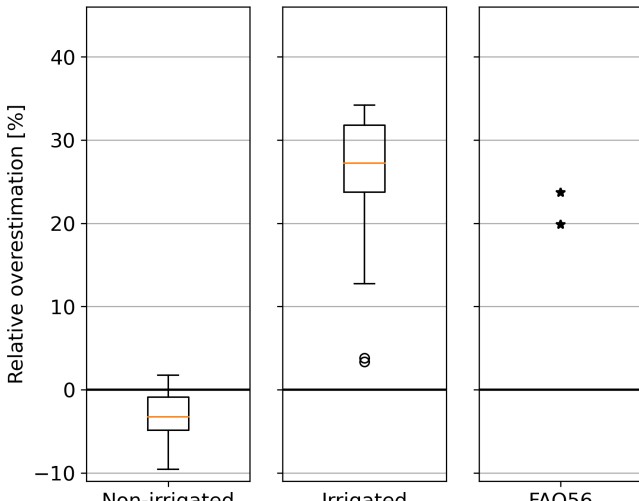

**Figure 8.** Mean relative overestimation of total evapotranspiration due to the hot, dry and windy atmospheric forcing file *atmo_NOIRR*, for different ISBA configurations, different irrigation parameterizations, and two irrigation-induced atmospheric effects of irrigation.

SOP. In the La Cendrosa alfalfa field, comparing the simulation with and without irrigation, it is shown that irrigation reduces
the 2-m air temperature by 2.6°C, increases the specific humidity by 1.8 g kg$^{-1}$ and reduces the wind speed by 0.7 m s$^{-1}$ on average during the SOP. These three irrigation-induced effects all contribute to a reduction in crop evapotranspiration. This is called atmospheric feedback on evapotranspiration.

The novelty of this work lies in the detailed analysis of the LSM processes involved in the atmospheric feedback on evapotranspiration, and in the quantification of this effect for different ISBA configurations. The different configurations studied
are commonly found in other LSMs and so it is presumed that the results presented herein are very likely to be valid for these LSMs as well. By focusing on specific days, the effect of atmospheric feedback on both evaporation and transpiration is studied: the atmospheric feedback is shown to substantially reduce evaporation and transpiration for well-irrigated crops. For water-stressed crops, however, other processes are involved. In such conditions, plants close their stomata and it is found that atmospheric feedback has a delaying effect on the timing of stomatal closure and an advancing effect on the timing of
stomatal reopening. This compensates for the decrease in transpiration or even leads to the opposite, i.e. an increase in transpiration due to the atmospheric feedback. It is also shown how transpiration and soil evaporation interact. Other things being equal, a decrease in transpiration leads to an increase in bare soil evaporation. This paper highlights and discusses how some parameterizations of ISBA affect these processes. In particular, the use of ISBA-A-$g_s$ increases the compensation effect of stomatal closure, and the use of the composite approach to vegetation representation increases the response of soil evaporation
to variation in transpiration.





Finally, the overestimation of evapotranspiration resulting from ignoring the atmospheric feedback over irrigated areas is quantified based on model results. It is found to be between 4% and 33%, with a mean of 25%. This overestimation is important and needs to be carefully considered for offline LSM simulations using atmospheric conditions unaffected by irrigation, i.e. most reanalyses and GCMs. This also has implications for impact studies under different future climate scenarios which make
estimates of future water needs for irrigation and the impact on plant phenology.

In fact, to the best of the authors' knowledge, no re-analysis has yet included irrigation in the LSM that is used in the coupling with the atmosphere. Some GCMs are starting to include it, but may miss some irrigation effects due to low resolution. However, the climate research community has recently initiated an international model intercomparison to improve understanding of the impact of irrigation on the Earth System (Yao et al., 2023). This intercomparison work will provide an opportunity to
assess the impact of irrigation in a more systematic way, and could promote the representation of irrigation in future GCM runs.

The present study further highlights the need for the inclusion of irrigation in any coupled surface–atmosphere model that has partial coverage of an irrigated area. This potentially simple addition to any LSM can lead to important improvements in weather modelling over semi-arid regions, and would also avoid important biases in evapotranspiration modelling for other
domains using the atmospheric outputs, particularly impact studies predicting the evolution of future water resources in regions such as the one studied here.

The present work relies on a fine-resolution coupled model to capture the irrigation effect on the atmosphere, and can subsequently quantify the importance of the atmospheric feedback in the offline LSM. As this coupled model has been widely used and studied for other aspects during the LIAISE SOP (see Lunel et al. (2024a) and Lunel et al. (2024b)), there is high
confidence in the quality and representativeness of the atmospheric forcings used here. However, it should be remembered that the LIAISE campaign SOP took place during a mostly sunny period, and the influence of irrigation on clouds could not be confidently demonstrated. Therefore the subsequent influence of a shortwave downward radiation modification on evapotranspiration was not explored in the current study. Since irrigation can influence boundary layer clouds in regions characterized by different topographies or for synoptic situations not considered herein (Kawase et al., 2008; Lobell et al.,
2008), this factor must be given special attention in other cases.

*Code and data availability.* The observational data sets analyzed in this study are available in the LIAISE database, accessible at *https://liaise.aeris-data.fr/page-catalogue/*. SURFEX is open-source and available at *http://www.umr-cnrm.fr/surfex/*. The generated model output data supporting the results of this study are available from the corresponding author upon reasonable request.

## Appendix A: Evapotranspiration units

Evapotranspiration values can be expressed in different units. In agronomy and hydrology the term evapotranspiration is often used and is given in mm h$^{-1}$ or mm day$^{-1}$. In meteorology the term latent heat flux $LE$ is more often used instead of



evapotranspiration and is given in W m$^{-2}$. However, evapotranspiration and latent heat flux are two ways of looking at the same water vapour flux. The two terms are linked by Eq. (A1) and can be used interchangeably since the latent heat of vaporization of water $L$ is often considered to be constant. Which term is used is more a matter of convention depending on the field, with meteorology tending to use latent heat flux and hydrology and agronomy tending to use evapotranspiration. In the present article the values of evapotranspiration are given both in terms of heat flux in W m$^{-2}$ and in terms of vapour flux in mm h$^{-1}$.

$$LE = \frac{L}{3600} \times ET \tag{A1}$$

where $LE$ is the latent heat flux in W m$^{-2}$, $L$ is the latent heat of vaporization of water and takes the value $2.26 \times 10^6$ J kg$^{-1}$, and $ET$ is the evapotranspiration in kg m$^{-2}$ h$^{-1}$ or mm h$^{-1}$.

**A1**

*Author contributions.* T. Lunel performed the simulation, processed the experimental and model data, performed the analysis and wrote the manuscript. B. Martí helped in setting up the simulations and in interpreting the results. A. Boone and P. Le Moigne supervised the research and helped in interpreting the results. All authors discussed the results and commented on the manuscript.

*Competing interests.* All authors report no competing interests.

*Acknowledgements.* The authors would like to gratefully acknowledge Daniel Martinez-Villagrasa and Guylaine Canut and their respective teams for providing observational data from the IRTA and La Cendrosa sites.



**Figure A1.** Evaporation modelled by ISBA for 18 (a, c) and 22 (b, d) July 2021 over a hypothetical dry parcel ($SWI$=0.2) located at the IRTA facility. The PFT modelled here corresponds to a drought-tolerant crop, with stomatal conductance modelled by ISBA-A-$g_s$. The vegetation is modelled with a composite approach for the upper panels (a, b) and with the explicit canopy approach (MEB) in the lower panels (c, d).





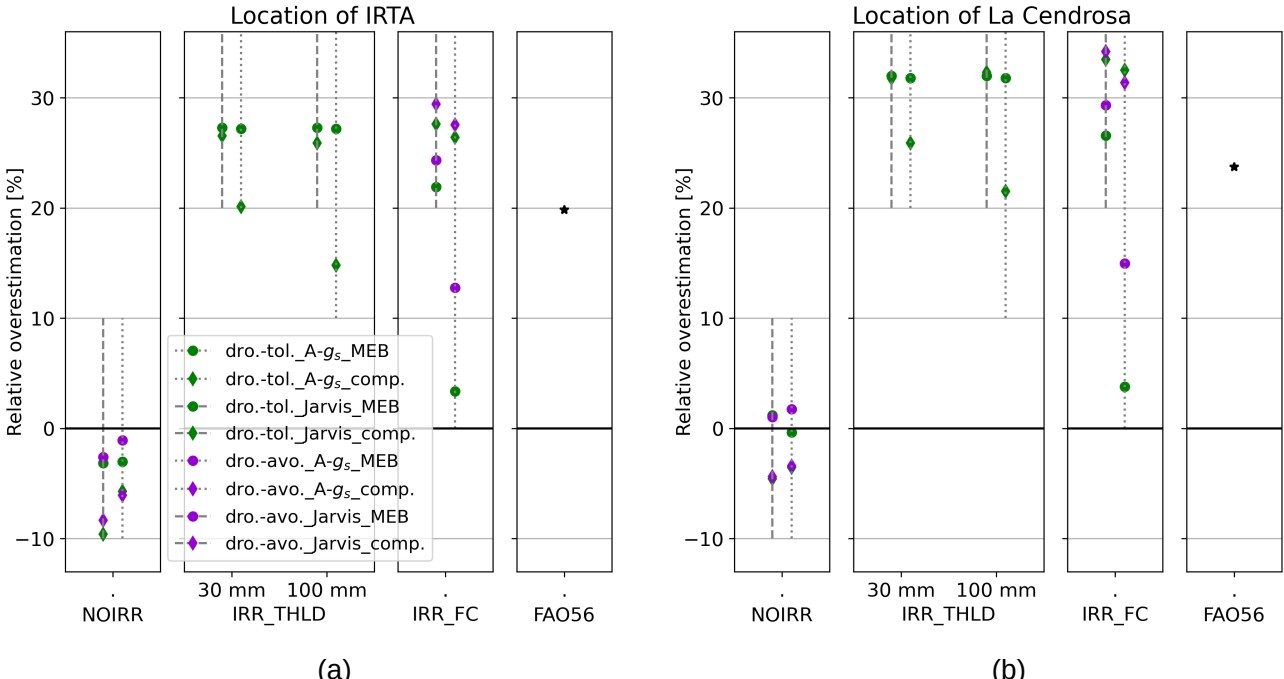

**Figure A2.** Mean relative overestimation of total evapotranspiration due to the hot, dry and windy atmospheric forcing *atmo_NOIRR*, for different ISBA configurations, different irrigation parameterizations, and two irrigation-induced atmospheric effects of irrigation, namely at La Cendrosa (a) and at the IRTA facility (b). Note that realistic parameterizations *IRR_THLD* are not applicable to drought-avoiding crops for technical reasons inherent to ISBA as available in SURFEX v8.1.

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
