# Peer review of "Systematic overestimation of evapotranspiration over irrigated areas by an offline land surface model"

_EGUsphere, 2024_

## Author Comment (AC1)

**Answer to RC2:**

*This is a well-written article on an important topic. The experimental design and data analysis seem adequate. I have only two very minor comments below. I couldn't find any visualization over the whole study duration. This can be beneficial to include.*

*Minor comments:*

*- Subsection numbering is not consistent. For instance, there is a "Materials and methods" section (Section #2), but no "Results" section.*

*- Figures 1-3: Is there any specific reason that the actual values over the entire experiment duration are not shown and only the average values are presented and compared with modeling schemes?*

Thank you for your very positive review. We did not provide visualisation over the whole study duration and opted for average values in Figures 1–3 to make the figures more readable. We will provide equivalent figures for the whole period in the supplementary materials of the revised version to avoid making the article too long. The section numbering will also be revised, and a 'Results' section will be added.

---

## Author Comment (AC2)

**Answers to RC1:**

Thank you for your thorough review. Your detailed feedback has been very helpful in clarifying and improving our manuscript. You will find the answers to each point below.

*This paper describes a sensitivity study of a Land Surface Model (ISBA) under two atmospheric forcing conditions using coupled and offline LSM-BLM/RCM over two irrigated plots in the LIASE campaign: one considering the feedback of irrigation on the atmospheric temperature and humidity profiles and one ignoring it. The description of the numerical experiment to generate both forcing datasets as well as their analysis in terms of evapotranspiration overestimation and other meteorological variables when ignoring irrigation are given in QJRMS ([https://doi.org/10.1002/qj.4736](https://doi.org/10.1002/qj.4736)).*

*The QJRMS article presents the direct effect of irrigation on evapotranspiration, i.e. how irrigation increases evapotranspiration from a given area, and its subsequent impact on the atmosphere. It is one of the papers used to validate atmospheric forcings. However, it does not discuss the indirect and second-order effects of irrigation on evapotranspiration, i.e. the atmospheric feedback on evapotranspiration, which leads to its overestimation in most offline LSMs. This is the topic of the present paper under review.*

*There are three main issues with the current manuscript:*

- *The importance of the impact of irrigation on the (changing) local/regional climate and the resulting areal evapotranspiration has been long recognized (cf. examples below); therefore, it is hard for the reader to identify the overall important message (to which it is referred lines 628-629), out of the obvious statement that irrigation affects the atmospheric conditions above the irrigated plot if the latter is large enough (wetter, cooler conditions) and should be considered when applying distributed hydrological or land surface models (statement which has been already put forward in the QJRMS paper) ?\**

The importance of atmospheric feedback on evapotranspiration (ET) has indeed already been recognised. However, its effect has rarely been accurately quantified for irrigated areas, and even less so in the context of offline LSM use. In LSMs, some complex ecophysiological behaviours influence modelled ET, resulting in potentially counterintuitive behaviour – like the increase of ET in response to the atmospheric feedback on some day due to the stomata control - which cannot be modelled using other approaches, such as the complementarity relationship (CR).

Important messages from this paper are, firstly, the quantification of ET overestimation in irrigated areas and, secondly, the absence of overestimation in non-irrigated areas when subject to the same atmospheric feedback. The quantification shows that when an offline LSM is run with atmospheric forcing that overlooks the effect of irrigation, modelled ET over an irrigated area may be overestimated by up to 35%. Secondly, over a water-stressed area, modelled ET will not be significantly overestimated. This effect is due to stomata closure, which is modelled by the ecophysiological schemes used in the LSM.

The distinct behaviours of ET in irrigated and water-stressed parcels subject to atmospheric feedback are entirely novel. Furthermore, while the quantification of atmospheric feedback on ET is not new, to the best of the authors knowledge, this is the first time it has been done using robust atmospheric data that so clearly discriminates the effect of irrigation, relying on already published articles for this part. This is also the first time that multiple configurations of an LSM, which are commonly used in other offline LSMs, have been tested, making the results particularly robust.

Another message from the present article is indeed to emphasize the need to consider the effect of irrigation in atmospheric forcings in distributed hydrological or offline LSMs. However, the currently available reanalyses do not contain this information, and some GCMs have only recently started to include it (see the IRRMIP exercise by Yao et al., 2025, in Nature), but only at low resolutions. Therefore, considering the effect of irrigation is not possible in most cases. Therefore, it is necessary to understand the value of the bias resulting from this simplification, which is provided by this work.

- *The sensitivity study is not generic enough to carry substantial interest for the HESS community (a few days only, with specific conditions, forcings are all locally generated, no benchmarks from a reanalysis have been used);*
  These are indeed weaknesses of the present article; however, they are counterparts to its strengths. Specifically, the effects of irrigation on the atmosphere are clearly demonstrated here thanks to the LIAISE special observation campaign and the numerous related articles (see Mangan et al., 2023a, QJRMS ; Brooke et al., 2024 ; Gonzalez-Armas, 2024 ; Udina et al, 2024 ; Lunel et al., 2024a and 2024b, available in the article bibliography), but this necessarily corresponds to a relatively short time period, i.e. the duration of the campaign. Also, exploring multiple LSM configurations and investigating the ecophysiological processes at play requires a focus on local case studies in order to keep the analysis digestible.
  The ERA5 reanalysis benchmark has already been performed for this period by Mangan et al., 2023b, AFM (see supp. bibliography below). This point will be added explicitly to the text.

  *[…] indeed, most LSM or hydrological model simulations can't be coupled to regional climate models and are therefore offline simulations (that is, the vast majority of the research presented in journals like HESS); it would have been interesting to provide a solution to bypass the bias induced by ignoring the feedback effect on the atmosphere by the irrigated areas:*
  Providing a solution to bypass the bias is indeed an interesting pathway for research, but is a wide and complex work that first needs a better understanding of the problem. The present paper therefore tries to advance our understanding of this issue, and is therefore important for guiding future work on potential bias correction. Specifically it gives quantified estimates of the bias that should be unbiased, but also it underlines the fact that such correction should consider differently well-watered and water-stressed areas submitted to the same atmospheric feedback.

*[…] Should one use a simple approach such as the Bouchet Complementarity for the subareas where irrigation takes place ? (cf. the ample literature by Jozsef Szilagyi to develop an equilibrium Priestly-Taylor formalism with a "wet bulb" like information ?);*

The Bouchet complementarity relationship is a research area that provides good evapotranspiration estimates at low cost and with limited data. However, it also has limitations. These limitations are either linked to the resolution of the input data (which is not applicable to high spatial or temporal resolutions (Szilagyi, 2018)) or to its inherent assumptions. The use of LSM is another line of research that increases our understanding of the water cycle as the key coupled processes are represented in a physically-based manner and it may bypass some limitations of other approaches, although it is not necessarily always better than the CR.

*[…] In the last decade several authors have tried to use the Bouchet Complementarity to analyse the effect of irrigation on regional climate change; I can mention at least one work in Turkey (Ozdogan and Salvucci, 2002) but also the "Australian Pan Evapotranspiration Decrease Controversy", for which several authors have analysed the cause of the decrease in pan ET data, first relating it to solar dimming (Roderick and Farquhar, 2002), but then realizing it is due mostly to the development of irrigation around the pans. I think that the literature review, which largely focuses on more recent works, should at least mention this line of research.*

These works are indeed interesting and will be mentioned in the revised version of the paper. However, they use a different approach based on the conceptual context of the complementarity relationship (CR). We believe that the current approach, which is based on the LSM, is distinct from the Bouchet CR and can therefore be considered complementary. Specifically, estimating the 'wet temperature' in Bouchet's CR is challenging and, in the present paper, is equivalent to evaluating the effect of irrigation on the atmosphere. We believe that the numerous articles on the LIAISE campaign render our estimates of the atmospheric impact of irrigation more robust than those obtained using the Priestley-Taylor approach. Ozdogan and Salvucci's work (2004) provides insightful views on the decrease of pan evapotranspiration in the context of an irrigated area under development. However, their methodology is very different from ours (based on observations across many years), as are their conclusions, which state that the decrease in pan evaporation is mainly due to a decrease in wind speed. The temperature effect is likely overlooked here because the temperature increase due to global warming compensates for the temperature decrease due to irrigation expansion, as seen in the Northwest Pacific by Lawston et al., 2020, Earth Interactions (cf article bibliography).

Our line of research based on LSMs does not focus much on potential or pan evaporation because the models do not require it. CR approaches do require potential ET and wet ET in order to calculate actual ET, whereas LSMs attempt to compute actual ET directly, bypassing some difficulties but creating others.

- *The paper lacks clarity at multiple places: for instance, the analysis of the impact on the partitioning between evaporation E and transpiration T for various soil moisture levels focuses on a common stress index, SWI, and it is not clear how it affects E and T separately; only E(SWI) is shown, and it is not sure whether in Fig. 7 the same SWI levels impacts E and T. In that case, why not focusing on different moisture controls for E and T ?*

The SWI is the main moisture control used for the transpiration and the evaporation in the ISBA LSM. This point will be mentioned more explicitly in the future revised version.

Each point in each subfigures of Fig. 7 correspond to an offline simulation with a constant SWI, meaning it is the same SWI levels that impact E and T. This point will be clarified in the revised version of the text. $\Delta E(SWI)$ is shown in Fig. 7 (b, d), and $\Delta T(SWI)$ is shown in Fig. 7 (a, c), with $\Delta$ refering to the mean absolute difference between the simulations without and with the atmospheric feedback. Absolute values E(SWI) and T(SWI) were not shown for the sake of brevity, but will now be added in Appendix.

The following text will be added l.557:

Soil moisture levels are represented by SWI because it is the main moisture control factor for transpiration in the ISBA model. Also evaporation is controlled by the ratio $Wg/Wfc$, which is conceptually similar to SWI (cf. Eq. 5). The SWI remains constant throughout each simulation, meaning that the same SWI levels impact transpiration and evaporation in Fig. 7.

*Also, what about advection from nearby drier areas in the simulation ? Is it dutifully accounted for (especially in those summer conditions) ?*

The advection from nearby areas is fully accounted for in the atmospheric forcing since it is represented explicitly in the mesoscale atmospheric model. This is actually a strength of our approach, since the atmospheric forcing is based on a fully coupled land–atmosphere system that has been validated against observations in two other papers (Lunel et al., 2024a, QJRMS; Lunel et al., 2024b, ACP). The first article focuses on the question of irrigation breezes, i.e. midday advection from irrigated areas towards dry areas (Lunel et al., 2024a, QJRMS), while the second article focuses on late afternoon marine air mass advection (Lunel et al., 2024b, ACP). The general findings of these articles are further validated on average over the LIAISE campaign in section 3 of the present paper.

*In summary, in order to make this work fully compliant with the readership of HESS, I would recommend extending the paper by offering solutions to assess and correct for the ET overestimation at regional and seasonal scales, using for example larger scale simulations (ERA5Land-like) on the LIAISE area. I don't really see what general outcomes one can draw from a few days of simulations over two plots, but maybe I missed out on an original sidelining interest of the paper.*

We acknowledge that our article has limitations and needs clarification on specific points, including the key findings, as mentioned above. However, we believe that our article provides a deeper

understanding of ET overestimation in irrigated areas by quantifying and highlighting processes that have not been discussed until now, such as stomatal control in water-stressed plants. This improved understanding is a step towards finding solutions to correct ET overestimation. However, it also shows that this correction might not be straightforward due to non-linear processes. Therefore, we will leave this research question for future projects.

In our opinion, the limited spatiotemporal coverage of our study is a counterpart of its strength. Specifically, this study builds upon various results of the LIAISE campaign that were obtained on the same limited spatiotemporal context. This short time period and small number of plots has allowed us to gain a fine understanding of the processes across various LSM configurations.

*Minor comments:*

*Line 33: "… is now fairly well understood and represented in LSMs": I would dampen this optimistic statement. It is still fairly hard to go below 30% error.*

Indeed, the sentence will be rewritten this way:
Modelling evapotranspiration in response to natural factors using LSMs is still an active area of research, and can produce satisfactory results depending on the context. However, this is not yet the case for responses to anthropogenic factors.

*Lines 138-144: This paragraph is curcial for the understanding of the paper, but lacks clarity and precision (e.g. "the surface features are not taken" etc).*

This paragraph will be rewritten as follows in the revised version:

*[N.B.: The reference to the coupled LSM-atmosphere simulations that were used to produce the atmospheric data has been removed to avoid confusing the reader. This is already discussed in section 2.4.]*

The land surface considered in the offline ISBA point scale simulations is that of a flat summer crop field. This land surface does not represent a particular field from the LIAISE campaign, but rather an average, typical Urgell region field. The sand and clay content is set at 33% each, making it a loamy clay soil. The Leaf Area Index (LAI) is set to 3 m2 m−2 in order to represent a well-established crop. The roughness length is set to 0.1 m, which is an average value for irrigated crops such as corn and alfalfa (Jacobs and Van Boxel, 1988; Otsuki et al., 1999).

*Line 240: how does SWI also potentially affect transpiration ?*

A few lines will be added here to clarify this point. In short this depends on the stomatal conductance scheme ; it is rather complicated with A-gs, but simpler with Jarvis where the effect is linear. The new appendix showing E(SWI) and T(SWI) (discussed above) will illustrate it in more details.

The following sentence will also be added to the text at line 244:

At wilting point the plant cannot extract water from the soil and transpiration drops to zero. At field capacity, the soil moisture does not limit the plant transpiration. The transpiration is then limited by other factors (incoming radiation, temperature, $CO_2$ concentration…).

*Lines 254-269 / 2.3: This echoes to my main comments 2 and 3: impact should be assessed at the seasonal scale, which is the scale of application of ET0; also, ET0 uses many assumptions that are only valid for a dense short grass fully irrigated, it is not meant to be considered in comparison with an output of an LSM outside of the numerous crop coefficient adjustments that are provided in the method to account for the "non standard" conditions (meteorological, moisture availability, type of the irrigation practice, plant development stage, etc), and typically not at the temporal scale of one single day.*

ET0 as defined with the Priestley-Taylor equation is indeed only fully relevant at the seasonal scale. However The ET0 of the FAO-56 (Allen et al., Crop evapotranspiration - Guidelines for computing crop water requirements, 1998) was actually designed to be used by farmers and in agricultural applications to assess the amount of irrigation water to apply on a dayly basis (cf chapter 2), or even hourly with an adapted version of the equation (cf chapter 4.1.2). The crop evapotranspiration ETc is then indeed found by multiplying ET0 with a crop coefficient Kc. This crop coefficient varies mainly with the growth stage of the crop and the regional climate zone, but is often assumed to be independent of the daily atmospheric conditions (Allen et al., 1998, Chapter 6). Therefore the absolute ETc overestimation due to the atmospheric feedback will be proportionnal to the ET0 overestimation, and the relative overestimation due to the atmospheric feedback will be the same.
Nevertheless we agree that ET0 estimates are fully valid under many assumptions. This comparison to FAO56-ET0 was done to make the link with simple methods for calculating ET, i.e. methods not relying on complex LSMs, but more on Penman-Monteith or Priestley-Taylor derived formulations, like done in some versions of the Bouchet complementarity relationship (Szilagyi, 2018).

*Line 566: it seems to me that it has been already recognized in the LSM community that moisture-driven stress factor in the Jarvis formulation is less realistic than one based on leaf water potential.*

Yes, however it is still widely used in many works, for the sake of simplicity and for the ease of interpretation going with it.

*Lines 570-578: here it would be helpful to have a discussion on what combinations of situations in the sensitivity are realistic in the given context (dry root zone vs wet surface soil moisture, no-irrig atmospheric conditions with high SWI etc etc).*

The following discussion will be added in the text at line 555.

Note that the combinations of soil moisture and different atmospheric conditions presented are realistic. Although irrigation has a significant impact on near-surface atmospheric conditions, synoptic conditions remain the primary driver of atmospheric properties. Consequently, it is possible to have a hot, dry atmosphere above a small irrigated area with wet soil, or conversely, a non-irrigated parcel with dry soil in the middle of an irrigated area with a relatively cool, humid atmosphere.

*Line 635: "a decrease in transpiration leads to an increase in bare soil evaporation": this automatically the case if using a coupled model such as MEB because any drop in transpiration (resp.*

*evaporation, for that matter) increases the temperature and decreases the humidity at the aerodynamic level and enhances evaporation (resp. transpiration); this is not a novelty, neither it is specific to those conditions.*

Indeed, this line was added for the sake of clarity for non-expert readers. We think it eases the understanding of the process at play, and therefore plan to keep this sentence in the revised version.

*Ozdogan, M., and G. D. Salvucci (2004), Irrigation-induced changes in potential evapotranspiration in southeastern Turkey: Test and application of Bouchet's complementary hypothesis, Water Resour. Res., 40, W04301, doi:10.1029/2003WR002822.*

*Roderick, M. L., and G. D. Farquhar (2002), The cause of decreased pan evaporation over the past 50 years, Science, 298, 1410– 1411.*

*\* Actually this is why a network of agrometeorological stations in standard conditions (i.e. well irrigated short grass) are usually required for computing reference evapotranspiration for monitoring plant water demand and plant water use in heavily irrigated areas, but rarely maintained in those standard conditions.*

Supplementary bibliography:

N.B.: The following bibliography will be added to the next version of the article, along with references to it where relevant in the text.

Mangan, M.-R., Hartogensis, O., Boone, A., Branch, O., Canut, G., Cuxart, J., Boer, H. J. D., Le Page, M., Martínez-villagrasa, D., Ramon, J., Price, J., & Vilà-guerau De Arellano, J. (2023). The surface-boundary layer connection across spatial scales of irrigation-driven thermal heterogeneity : An integrated data and modeling study of the LIAISE field campaign. *Agricultural and Forest Meteorology*, *335*(April), 109452. https://doi.org/10.1016/j.agrformet.2023.109452

Szilagyi, Jozsef; A calibration-free, robust estimation of monthly land surface evapotranspiration rates for continental-scale hydrology. *Hydrology Research*. 2018; 49 (3): 648–657. doi: https://doi.org/10.2166/nh.2017.078

Yao, Y., Ducharne, A., Cook, B.I. *et al.* Impacts of irrigation expansion on moist-heat stress based on IRRMIP results. *Nat Commun* **16**, 1045 (2025). https://doi.org/10.1038/s41467-025-56356-1

---

## Author Comment (AC3)

**Answer to CC1:**

*This paper presents a robust and insightful analysis of the systematic overestimation of evapotranspiration over irrigated areas by offline land surface models. The authors effectively integrate field observations with advanced coupled modeling techniques to dissect the complex interplay between irrigation-induced atmospheric changes and land surface processes. Their detailed examination of various ISBA configurations, combined with a thorough validation against observational data, not only deepens our understanding of the atmospheric feedback mechanisms but also offers valuable guidance for improving model performance in both weather forecasting and water resource management. The comprehensive approach and meticulous quantification of key processes make this work a significant contribution to the field.*

Thank you for your very positive review. Below are the answers to your specific remarks and questions.

- *How sensitive are the results to the various ISBA configuration choices (e.g., canopy representation, stomatal conductance schemes, drought response) and to what extent might these choices limit the generalizability of the findings to other LSM frameworks?*

   The answer is partly provided in Appendix A2 of the Appendices. The relative overestimation of irrigated parcels does not depend much on the configuration, except for some combinations, such as when the MEB and A-gs schemes are activated together. In this case, the overestimation ranges from 3 to 15%, as shown in Figure A2.

- *Given that the atmospheric forcings (atmo_NOIRR and atmo_IRR_FC) are derived from a specific coupled model simulation over the LIAISE campaign period, how representative are these forcings for other irrigated regions or different meteorological conditions?*

   This is indeed a limitation of our article. The forcings are representative of a cold, semi-arid climate (Bsk in the Köppen-Geiger classification), and the effect of irrigation on the atmosphere is representative of a large, densely irrigated area. For different climate conditions and irrigated areas, the overestimation values may differ, and further research is needed. However, Decker et al. (2017) showed similar values of ET overestimation for a less densely irrigated region. Therefore, our article suggests that the value of 25% overestimation is somewhat generalisable to different topographies of irrigated regions. This limitation will be clarified in the discussion section.

- *The paper relies on validation using data from two field sites—how robust is the model evaluation across diverse settings, and what uncertainties remain in the comparison between modeled and observed near-surface meteorological variables?*

   We believe that the diverse settings used are representative of those often used in other LSMs and that impact ET the most, so the evaluation is quite robust in our opinion.

   However, some uncertainties remain concerning the differences between modelled and observed near-surface meteorological variables. Coupled models are not yet perfect, and the modelled

variables do not always align closely with observations. In this study, it is considered that the difference between the irrigated atmospheric forcing and the observations stems from shortcomings in the coupled model, rather than missing irrigation effects. However, this cannot be verified. Therefore, the uncertainty is essentially similar to that obtained with mesoscale land-atmosphere coupled models.

- *Can the authors clarify how the compensatory interactions between transpiration and soil evaporation are quantified, and what are the uncertainties associated with isolating the atmospheric feedback effects on these individual processes?*

The interactions between transpiration and evaporation are analysed by investigating their respective behaviours in each simulation. Figure 7 shows the atmospheric feedback on transpiration and evaporation separately, but this does not mean that the effects are isolated. Actually in LSMs transpiration and evaporation cannot be considered purely individual processes since they interact at least via the surface temperature. In fact, the behaviours shown in Fig. 7(a) and (b) (or (c) and (d)) must be interpreted together to understand the interactions between the two components. This point will be clarified in the revised version.

New text:
The lower the transpiration, the higher the top soil temperature and ultimately the higher the evaporation, i.e. evaporation compensates, to a certain degree, for the lack of transpiration.

l.585: The interaction between transpiration and evaporation through surface temperature (either canopy or composite skin temperature) means these two processes are necessarily intertwined and must be interpreted together.

- *How might the biases associated with the offline LSM approach (due to missing irrigation-induced atmospheric feedback) impact downstream applications in water resource management and agricultural planning, and what strategies are proposed to mitigate these limitations in operational settings?*

The presented biases can directly impact agricultural or water management applications. For example, overestimating evapotranspiration (ET) can lead to recommendations for irrigation water amounts being overestimated, resulting in excessive water consumption. There are numerous potential strategies to mitigate these biases, and determining the most effective strategy is another topic for research. The best option would probably be to use atmospheric forcings that represent the effect of irrigation on the atmosphere. However, this is rare, and obtaining the corresponding atmospheric data is computationally expensive. One possible simplistic approach would be to decrease ET estimates by 25% above irrigated parcels, but not above non-irrigated parcels. Another option would be to apply a negative uncertainty of 30% to ET estimates above irrigated parcels. We therefore prefer to leave this question open and allow readers to use whichever approach they prefer.

*Additionally, it would strengthen the manuscript to reference recent advances in remote sensing applications in hydrological modeling. In particular, please consider citing the paper 'Assimilation of Sentinel-based Leaf Area Index for Modeling Surface-Groundwater Interactions in Irrigation Districts'*

*to provide further context and support for the integration of satellite-based vegetation parameters in modeling surface–groundwater interactions in irrigated areas.*

Thank you for your proposal. Although we find remote sensing applications very interesting and promising for hydrological modelling, we feel that the present article is only weakly related to them. Therefore, we prefer to keep the article concise.